# Targeted degradation of BRD9 reverses oncogenic gene expression in synovial sarcoma

Gerard L Brien[1,2]*, David Remillard[3,4], Junwei Shi[5], Matthew L Hemming[1,3], Jonathon Chabon[1], Kieran Wynne[6], Eugène T Dillon[6], Gerard Cagney[6], Guido Van Mierlo[7], Marijke P Baltissen[7], Michiel Vermeulen[7], Jun Qi[4], Stefan Fröhling[8,9,10], Nathanael S Gray[4], James E Bradner[3†], Christopher R Vakoc[11], Scott A Armstrong[1]*

[1]Department of Pediatric Oncology, Dana Farber Cancer Institute, Boston Children's Hospital and Harvard Medical School, Boston, United States; [2]Smurfit Institute of Genetics, Trinity College Dublin, Dublin, Ireland; [3]Department of Medical Oncology, Dana Farber Cancer Institute, Boston Children's Hospital and Harvard Medical School, Boston, United States; [4]Department of Cancer Biology, Dana Farber Cancer Institute, Boston Children's Hospital and Harvard Medical School, Boston, United States; [5]Department of Cancer Biology, Abramson Family Cancer Research Institute, Perelman School of Medicine, University of Pennsylvania, Philadelphia, United States; [6]School of Biomolecular and Biomedical Science and Conway Institute, University College Dublin, Dublin, Ireland; [7]Department of Molecular Biology, Faculty of Science, Radboud Institute for Molecular Life Sciences, Oncode Institute, Radboud University Nijmegen, Nijmegen, The Netherlands; [8]German Cancer Consortium, Heidelberg, Germany; [9]Section for Personalized Oncology, Heidelberg University Hospital, Heidelberg, Germany; [10]Division of Translational Oncology, National Center for Tumor Diseases Heidelberg and German Cancer Research Center, Heidelberg, Germany; [11]Cold Spring Harbor Laboratory, Cold Spring Harbor, United States

*For correspondence:
gbrien@tcd.ie (GLB);
Scott_armstrong@dfci.harvard.edu (SAA)

Present address: †Novartis Institutes for BioMedical Research, Cambridge, United States

**Abstract** Synovial sarcoma tumours contain a characteristic fusion protein, SS18-SSX, which drives disease development. Targeting oncogenic fusion proteins presents an attractive therapeutic opportunity. However, SS18-SSX has proven intractable for therapeutic intervention. Using a domain-focused CRISPR screen we identified the bromodomain of BRD9 as a critical functional dependency in synovial sarcoma. BRD9 is a component of SS18-SSX containing BAF complexes in synovial sarcoma cells; and integration of BRD9 into these complexes is critical for cell growth. Moreover BRD9 and SS18-SSX co-localize extensively on the synovial sarcoma genome. Remarkably, synovial sarcoma cells are highly sensitive to a novel small molecule degrader of BRD9, while other sarcoma subtypes are unaffected. Degradation of BRD9 induces downregulation of oncogenic transcriptional programs and inhibits tumour progression in vivo. We demonstrate that BRD9 supports oncogenic mechanisms underlying the SS18-SSX fusion in synovial sarcoma and highlight targeted degradation of BRD9 as a potential therapeutic opportunity in this disease.
DOI: https://doi.org/10.7554/eLife.41305.001

## Introduction

Sarcomas although rare in adult patients account for up to 20% of all paediatric malignancies (*Burningham et al., 2012*). These are often aggressive diseases which do not respond well to conventional therapeutic interventions (*Anderson et al., 2012*). As such the cure rates for many of these diseases are unsatisfactory and patient prognoses remain poor. The molecular pathology of many of these cancers is associated with recurrent chromosomal rearrangements; leading to the generation of chimeric fusion proteins. Significantly, many fusion protein generating aberrations occur in a genomic background with few co-occurring genetic alterations (*Gao et al., 2018*; *Brohl et al., 2014*; *Tirode et al., 2014*; *Crompton et al., 2014*; *Seki et al., 2015*; *Shern et al., 2014*). This has led to the prevailing notion that these gene fusions are often the primary driver of disease development. These chromosomal rearrangements often effect genes involved in transcriptional/chromatin regulatory mechanisms; with the resulting fusion proteins thought to drive disease development by altering the dynamics of transcriptional control. Excitingly, recent work has highlighted the therapeutic potential of targeting mechanisms of transcriptional control in cancer cells (*Brien et al., 2016*). However, effective means of blocking oncogenic transcriptional mechanisms in fusion gene driven sarcomas are currently lacking.

Synovial sarcoma is a fusion gene driven malignancy, which accounts for ~10% of soft-tissue sarcomas. Synovial sarcoma is a poorly differentiated malignancy with an often aggressive clinical progression. It occurs in patients of all ages, but is particularly common in children and young adults with a peak incidence between 20–30 years of age. The hallmark genetic abnormality in synovial sarcoma is a recurrent t(X;18) chromosomal rearrangement. This fuses the *SS18* gene (also known as *SYT*) on chromosome 18 to one of three related genes *SSX1*, *SSX2* and *SSX4* on the X chromosome (*Clark et al., 1994*; *de Leeuw et al., 1995*; *Skytting, 1999*). This fusion is considered pathognomonic for the disease, with diagnoses confirmed by RT-PCR and karyotyping analyses to identify the fusion event. As such, essentially 100% of synovial sarcoma tumours contain an SS18-SSX fusion. The SS18-SSX rearrangement is often the only genetic abnormality in synovial sarcoma tumours (*Cancer Genome Atlas Research Network, 2017*; *Barretina et al., 2010*); suggesting that it is the primary driver of disease. Indeed, conditional expression of SS18-SSX in muscle progenitor cells leads to development of a fully penetrant synovial sarcoma like disease in mice (*Haldar et al., 2007*).

The SS18-SSX fusion protein is believed to function as an aberrant transcriptional regulator. The SS18 protein is a dedicated component of the chromatin remodelling BAF (also known as SWI/SNF) complex which functions primarily in transcriptional activation (*Middeljans et al., 2012*; *Kadoch and Crabtree, 2013*). Whereas the SSX proteins are thought to function in gene silencing; potentially through interactions with the Polycomb Repressive Complex (PRC)1 (*Lim et al., 1998*; *Soulez et al., 1999*). SS18-SSX dominantly assembles into BAF complexes in synovial sarcoma cells, leading to eviction of the wildtype SS18 and SNF5 proteins from the complex. This altered complex assembly is redistributed on chromatin and drives an expression signature required to maintain the proliferative/undifferentiated state of synovial sarcoma cells (*McBride et al., 2018*; *Banito et al., 2018*). SS18-SSX chromatin binding is directed in part through interactions with the PRC1.1 complex; mediated by the SSX portion of the fusion (*Banito et al., 2018*). The recruitment of SS18-SSX to chromatin via interactions with PRC1.1 is essential for the oncogenic function of the fusion. Moreover, association of SS18-SSX with DNA-binding transcription factors has also been suggested to be important for chromatin binding and oncogenic activities (*Su et al., 2012*). Recruitment of BAF complex activity to SS18-SSX bound regions is essential for transcriptional activation of fusion target genes (*McBride et al., 2018*; *Banito et al., 2018*; *Kadoch et al., 2017*). Depletion of SS18-SSX protein levels leads to reduced BAF complex binding at target sites and repression of fusion target genes. These findings highlight that SS18-SSX driven alterations in chromatin regulatory pathways are a key aspect of synovial sarcoma oncogenesis. Moreover, they highlight that targeting mechanisms related to fusion protein recruitment and BAF complex function may provide a therapeutic opportunity in this disease. However, to date robust approaches for targeting these mechanisms have not been described.

Here using a custom CRISPR/Cas9 based functional genomics approach focused on chromatin regulatory genes we identify the bromodomain of BRD9 as a vulnerability in synovial sarcoma cells. We show that BRD9 is part of SS18-SSX containing BAF complexes in synovial sarcoma cells; and

that the association of BRD9 with the BAF complex is functionally essential. Targeting BRD9 with a novel chemical degrader specifically impedes synovial sarcoma cell viability; eliciting more robust therapeutic effects than BRD9 inhibition using bromodomain targeting chemical probes. Importantly, BRD9 is required to maintain appropriate expression of an oncogenic gene expression signature driven by SS18-SSX. Taken together, our findings highlight BRD9 as a novel therapeutic target in synovial sarcoma.

## Results

### The BRD9 bromodomain is a functional dependency in synovial sarcoma

To identify functional chromatin based dependencies that may be amenable to therapeutic targeting in synovial sarcoma cells we used a CRISPR/Cas9 based domain focused pooled screening approach (*Shi et al., 2015*). To this end we generated a custom lentiviral sgRNA library targeting known functional regions in 193 chromatin regulatory proteins. Viral supernatants generated with this library were used to infect Cas9 expressing synovial and Ewing sarcoma cell lines. The relative abundance of individual sgRNAs within each population was compared between early and late time points by high-throughput sequencing (*Figure 1A*). These analyses demonstrated that three independent sgRNAs targeting the bromodomain of BRD9 were depleted from synovial, but not Ewing sarcoma cell cultures (*Figure 1B–C*, and *Figure 1—source datas 1* and *2*). Remarkably, of the 52 bromodomains contained within 38 proteins targeted in this library, only the BRD9 bromodomain had all sgRNAs specifically depleted in synovial sarcoma cells (*Figure 1—figure supplement 1A*). This is in striking contrast to the bromodomains of BRD4 which are a dependency in both synovial and Ewing sarcoma cells, as well as several other malignancies (*Shi and Vakoc, 2014*; *Hensel et al., 2016*). To further examine the specificity of this dependency we performed individual sgRNA depletion assays in two independent synovial, Ewing and rhabdomyosarcoma cell lines, respectively. These experiments demonstrated that BRD9 bromodomain targeting sgRNAs were only depleted in synovial sarcoma cells (*Figure 1D* and *Figure 1—source data 3*). Importantly, the sgRNAs used here have comparable or higher genome editing efficiencies in Ewing and rhabdomyosarcoma cell lines, compared to synovial sarcoma cells (*Figure 1—figure supplement 1B* and data not shown). This indicates that differences in sgRNA depletion cannot be attributed to discrepancies in sgRNA editing. Moreover, BRD9 expression levels are consistent across all cell lines tested, indicating that differences in BRD9 levels do not reflect altered sensitivity to BRD9 targeting (*Figure 1—figure supplement 1C*). Using an independent shRNA-based approach we observed similar synovial sarcoma specific effects following knockdown of BRD9 protein levels (*Figure 1—figure supplement 1D–G*). Consistent with this, within the recently published Project DRIVE (*McDonald et al., 2017*) database we observe, that among the almost 400 cancer cell lines assayed, synovial sarcoma cell lines are the most sensitive to BRD9 targeting (*Figure 1E*). To confirm the importance of the BRD9 bromodomain we performed functional rescue experiments. We generated a full-length human BRD9 cDNA containing silent point mutations within the sgRNA recognition sequence, conferring resistance to Cas9 targeting (*Figure 1—figure supplement 1H*). Next, we expressed a full-length (FL), bromodomain deleted (Δbromo) or bromodomain inactivated (N216A) version of this cDNA in synovial sarcoma cells. This demonstrated that only wildtype full-length BRD9 can rescue the depletion of sgRNAs targeting the bromodomain (*Figure 1F*, *Figure 1—source data 4*, *Figure 1—figure supplement 1I*). This indicates that BRD9, and the BRD9 bromodomain, are selective functional dependencies in synovial sarcoma; highlighting a novel therapeutic target in this disease.

### BRD9 is a component of SS18-SSX containing BAF complexes

BRD9 has previously been shown to be a component of the BAF complex in several normal tissues (*Kadoch et al., 2013*). Moreover, biochemical studies in HEK293T cells have indicated that BRD9 can also associate with SS18-SSX containing complexes in this setting (*Middeljans et al., 2012*). However, it is unknown whether BRD9 is part of the oncogenic SS18-SSX containing BAF complex in synovial sarcoma cells. Consistent with previous results we found that BRD9 associates with exogenously expressed SS18-SSX1/2 in HEK293T cells (*Figure 2—figure supplement 1A–C*,*Figure 2*-figure supplement-source data 1–3). Next, to test whether SS18-SSX fusions also interact with BRD9 in synovial sarcoma cells, we immunoprecipitated the endogenous fusion protein in two independent

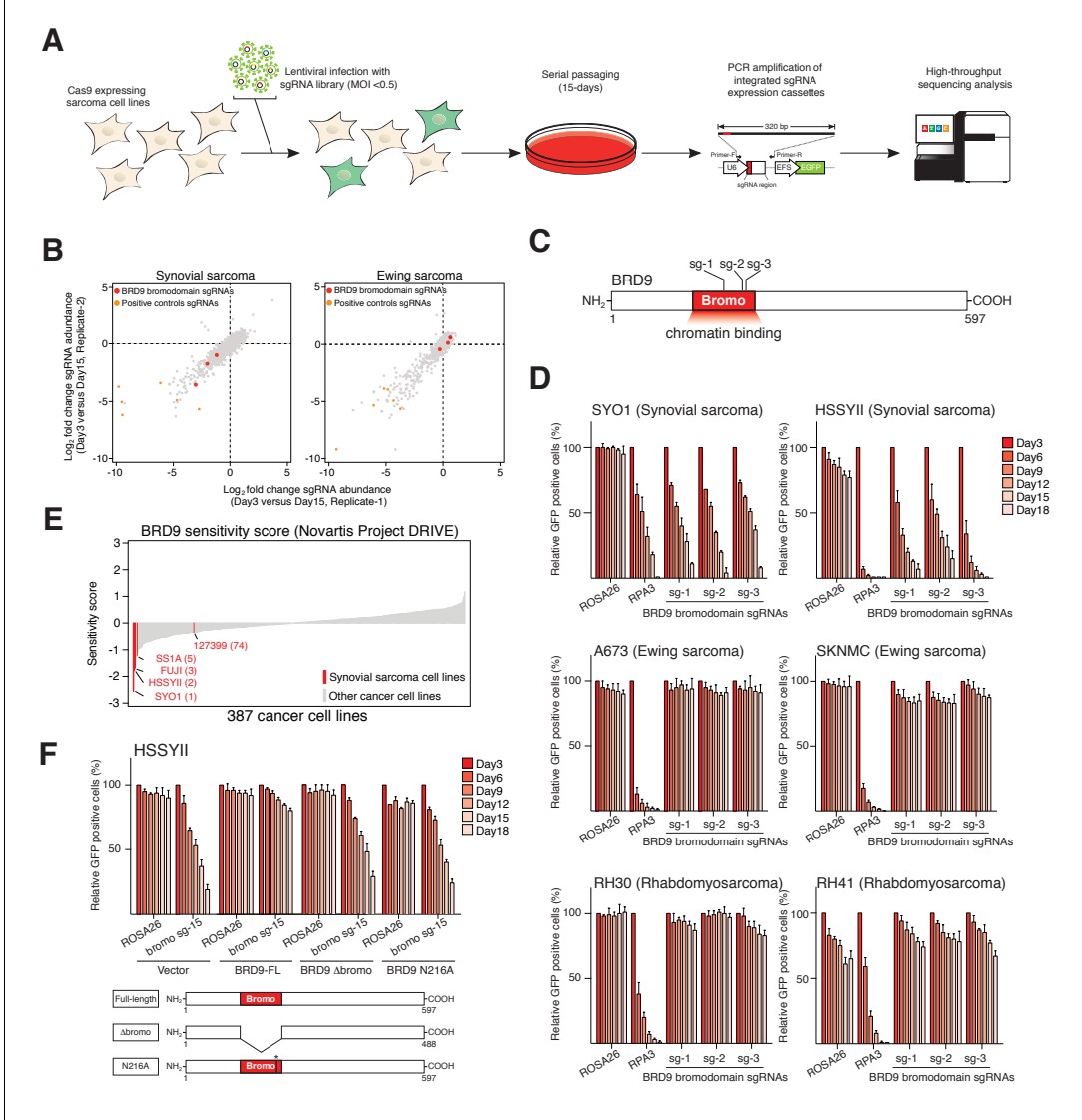

**Figure 1.** The BRD9 bromodomain is a functional dependency in synovial sarcoma. (A) Schematic representation of CRISPR/Cas9 based genomic screening approach. (B) Scatter plot representation of biological duplicate sgRNA screening data in synovial and Ewing's sarcoma cell lines. Each dot denotes an individual sgRNA and axes represent log₂ fold-change in sgRNA abundance between day-3 and day-15. BRD9 bromodomain and control sgRNAs are highlighted. (C) Schematic representation of the BRD9 protein structure with sgRNA target sites indicated. (D) Negative selection based CRISPR-Cas9 mutagenesis assays. The relative GFP+ (sgRNA+) subpopulation percentage is depicted at the indicated time-points after lentiviral infection. Mean ± s.d., n = 3. (E Waterfall plot representing 'BRD9 sensitivity' score in a panel of cancer cell lines taken from the Project DRIVE database (ref. *Lim et al., 1998*) (https://oncologynibr.shinyapps.io/drive/). (F) Negative selection based CRISPR-Cas9 mutagenesis assays in bromodomain functional rescue experiments. The relative GFP+ (sgRNA+) subpopulation percentage is depicted at the indicated time-points after lentiviral infection. Mean ± s.d., n = 3.

DOI: https://doi.org/10.7554/eLife.41305.002

The following source data and figure supplement are available for figure 1:

**Source data 1.** Sequencing read counts and fold-change values for individual sgRNAs in library experiments in HSSYII synovial sarcoma cells.
DOI: https://doi.org/10.7554/eLife.41305.004
**Source data 2.** Sequencing read counts and fold-change values for individual sgRNAs in library experiments in A673 Ewing sarcoma cells.
DOI: https://doi.org/10.7554/eLife.41305.005
**Source data 3.** Relative GFP positive percentages in negative selection sgRNA assays in six independent pediatric sarcoma cell lines.
DOI: https://doi.org/10.7554/eLife.41305.006
**Source data 4.** Relative GFP positive percentages in negative selection sgRNA assays in BRD9-FL, BRD9-Dbromo or BRD9-N216A rescue experiments performed in HSSYII cells.

*Figure 1 continued on next page*

*Figure 1 continued*

DOI: https://doi.org/10.7554/eLife.41305.007

**Figure supplement 1.** BRD9 is a specific functional dependency in synovial sarcoma.

DOI: https://doi.org/10.7554/eLife.41305.003

synovial sarcoma cell lines (*Figure 2A*). Significantly, these experiments demonstrated that BRD9 co-purifies with endogenous SS18-SSX containing BAF complexes in synovial sarcoma cells (*Figure 2B*, *Figure 2—source data 1* and *2*, *Figure 2—figure supplement 1D* and *Figure 2—figure supplement 1—source data 4*). Moreover, SS18-SSX fusion proteins co-purify with BRD9 in reciprocal endogenous IP experiments (*Figure 2C and D*). A recent report indicates that BRD9 is a member of a novel subclass of BAF complex(es), termed GBAF (for GLTSCR1/1L-BAF) (*Alpsoy and Dykhuizen, 2018*). These complexes lack SNF5 and an ARID component and contain BRD9, GLTSCR1 or GLTSCR1L as defining complex members. Our proteomic analysis of endogenous SS18-SSX containing complexes identified peptides mapping to GLTSCR1, lending support to the notion that the fusion incorporates into GBAF assemblies containing BRD9 (*Figure 2—figure supplement 1C and D*). As such by combining genomic and proteomic approaches we have identified BRD9 as a functional dependency within SS18-SSX fusion protein containing BAF complexes in synovial sarcoma cells.

To ascertain the relative proportion of individual BAF complex members in SS18-SSX purifications we used the intensity-based absolute quantification (iBAQ) algorithim (*Schwanhäusser et al., 2011*). This showed that core complex members such as SMARCC1, SMARCC2 and SMARCA4 have relative abundances approximately equal to, or greater than, SS18-SSX (*Figure 2E*); suggesting that these proteins co-exist with the fusion protein in most (if not all) complexes. However, the relative abundance of BRD9 (and GLTSCR1) is 10–20% that of SS18-SSX; indicating that these components are sub-stoichiometric members of SS18-SSX containing complexes. Interestingly, several of the BAF complex members (PBRM, SMARCA2 and SMARCA4) identified in these proteomics studies were included in our functional genomics screen (*Figure 1A*). However, no robust dependencies were evident among the bromodomains of these proteins which were targeted within our library (*Figure 2—figure supplement 1E*). Intriguingly, this suggests that the minor subset of BRD9 containing complexes are particularly important, and perhaps functionally specialised in synovial sarcoma cells.

## BRD9 functions within SS18-SSX containing complexes

Next, we wanted to understand whether BRD9 executes any bromodomain independent functions in synovial sarcoma cells. To do this we used a high-density CRISPR mutagenesis approach, introducing 92 individual sgRNAs targeting across the BRD9 locus into Cas9 expressing synovial sarcoma cell lines. We monitored for changes in sgRNA expressing (GFP-positive) cells over time, and consistent with our pooled screen most sgRNAs targeting the BRD9 bromodomain were robustly out competed in these GFP depletion assays (*Figure 2F* and *Figure 2—source data 3*). However, we identified an additional hotspot of sgRNA depletion within a previously uncharacterised central region of BRD9 (amino acids 311–345). We confirmed the importance of this region with functional rescue experiments showing that a Δ311–345 BRD9 cDNA was incapable of rescuing the depletion of sgRNAs targeting this region (*Figure 2G* and *Figure 2—source data 4*). Strikingly, co-IP experiments found that while the BRD9 bromodomain is dispensable for BAF complex interaction, this novel functional region is essential for association with the complex (*Figure 2H*). These results identify a novel BAF complex interaction domain within BRD9 and demonstrate that association of BRD9 with the BAF complex is functionally essential in synovial sarcoma cells.

## BRD9 co-binds the synovial sarcoma genome with SS18-SSX

To understand the extent to which BRD9 and SS18-SSX containing complexes overlap on chromatin we performed chromatin immunoprecipitation with next-generation sequencing (ChIP-seq). Owing to a lack of high-quality ChIP-grade antibodies for BRD9 and SS18-SSX, we adapted a previously reported CRISPR/Cas9 based approach to knock-in a 3xHA epitope tag at the C-termini of the endogenous *BRD9* and *SS18-SSX1* loci in HSSYII cells (*Savic et al., 2015*) (*Figure 3—figure supplement 1A*). ChIP-seq analyses demonstrated that BRD9 and SS18-SSX1 bind broadly throughout the

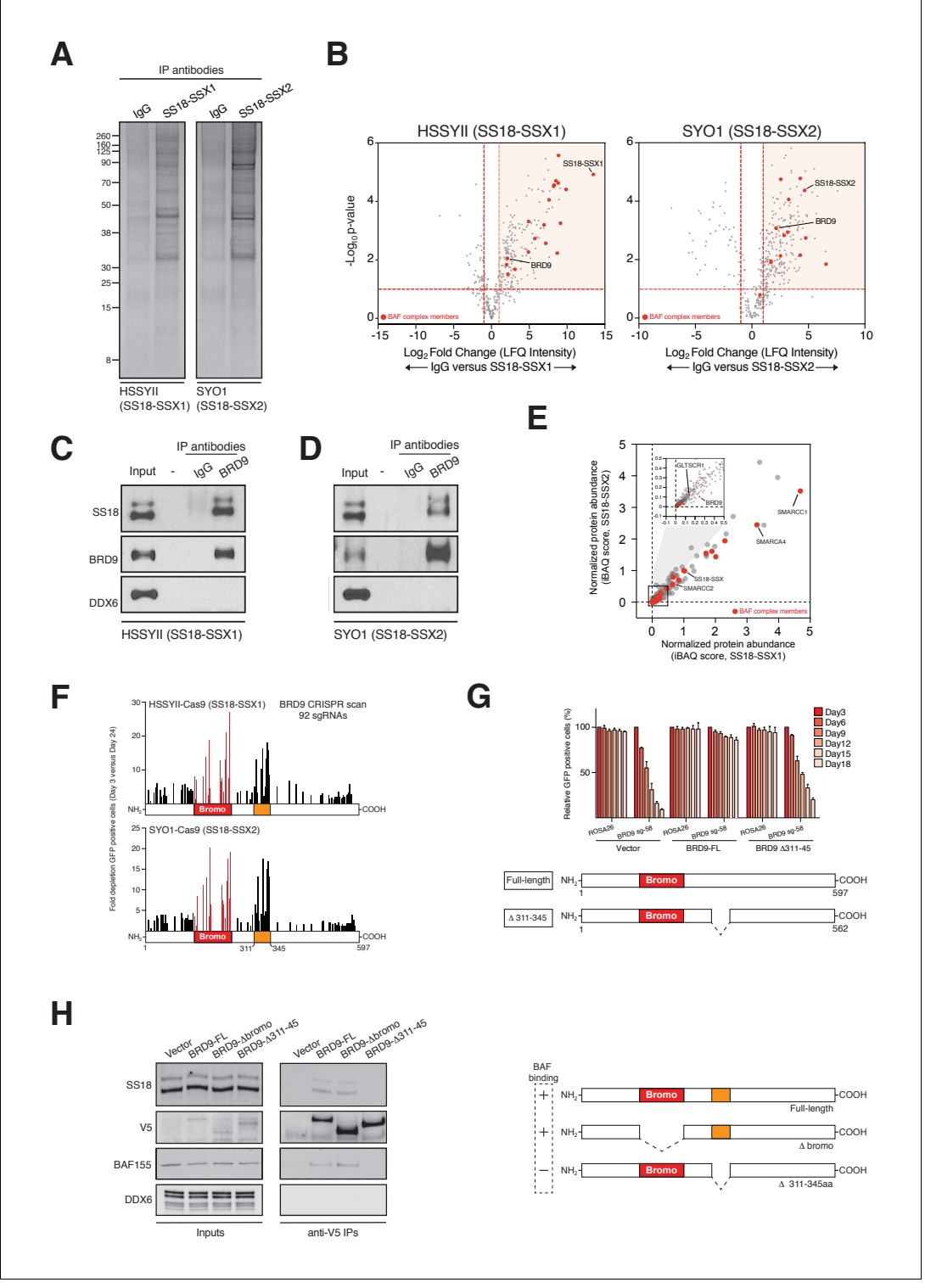

**Figure 2.** BRD9 functions as part of SS18-SSX containing SWI/SNF complexes. (**A**) Silver stains of endogenous SS18-SSX fusion protein immunoprecipitations performed on nuclear protein lysates prepared two independent synovial sarcoma cell lines. (**B**) Volcano plots representing fold enrichment (LFQ intensity) of proteins identified by mass spec in SS18-SSX1 or SS18-SSX2 purifications relative to IgG control purifications. Known BAF members are indicated in red. (**C**) Western blots analyses of the indicated proteins performed on endogenous BRD9 or IgG purifications in HSSYII cells (Input = 10% total IP material). (**D**) Western blots analyses of the indicated proteins performed on endogenous BRD9 or IgG purifications in SYO1 cells (Input = 10% total IP material). (**E**) Scatter plot

*Figure 2 continued on next page*

*Figure 2 continued*

representing the normalized protein abundance (IBAQ score) of proteins identified in SS18-SSX1 and SS18-SSX2 purifications. Known BAF members are indicated in red (F) High density sgRNA tiling of BRD9 in two independent SScell lines. Each bar represents the fold-change of an individual sgRNA and its target site along the BRD9 protein. (G) Negative selection based CRISPR-Cas9 mutagenesis assays in amino acid 311–345 region functional rescue experiments. The relative GFP[+] (sgRNA[+]) subpopulation percentage is depicted at the indicated time-points after lentiviral infection. Mean ± s.d., n = 3. (H) Western blot analyses of the indicated proteins in anti-V5 purifications performed in control HSSYII cells, or HSSYII cells expressing a full-length, bromodomain deleted or amino acid 311–345 deleted BRD9.
DOI: https://doi.org/10.7554/eLife.41305.008

The following source data and figure supplements are available for figure 2:

**Source data 1.** Mass spectrometry data from endogenous SS18-SSX1 purifications in HSSYII cells.
DOI: https://doi.org/10.7554/eLife.41305.014
**Source data 2.** Mass spectrometry data from endogenous SS18-SSX2 purifications in SYO1 cells.
DOI: https://doi.org/10.7554/eLife.41305.015
**Source data 3.** Fold depletion of GFP positive cells in negative selections sgRNA assays in HSSYII and SYO1 cells in BRD9 sgRNA tiling experiments.
DOI: https://doi.org/10.7554/eLife.41305.016
**Source data 4.** Relative GFP positive percentages in negative selection sgRNA assays in BRD9-FL, BRD9-D311-345 rescue experiments performed in HSSYII cells.
DOI: https://doi.org/10.7554/eLife.41305.017
**Figure supplement 1.** BRD9 is a component of SS18-SSX containing BAF complexes.
DOI: https://doi.org/10.7554/eLife.41305.009
**Figure supplement 1—source data 1.** Mass spectrometry data from SS18-SSX1 purifications in HEK293T cells.
DOI: https://doi.org/10.7554/eLife.41305.010
**Figure supplement 1—source data 2.** Mass spectrometry data from SS18-SSX2 purifications in HEK293T cells.
DOI: https://doi.org/10.7554/eLife.41305.011
**Figure supplement 1—source data 3.** Presented is the number of peptides mapping to each of the indicated BAF complex members in purifications of HA-tagged SS18-SSX1 and SS18-SSX2 expressed in HEK293T cells.
DOI: https://doi.org/10.7554/eLife.41305.012
**Figure supplement 1—source data 4.** Presented is the number of peptides mapping to each of the indicated BAF complex members in purifications of endogenous SS18-SSX1 and SS18-SSX2 expressed in HSSYII and SYO1 cells.
DOI: https://doi.org/10.7554/eLife.41305.013

genome (*Figure 3A*); with ~35% of binding sites occurring at gene promoters and the remaining ~65% at distal inter- and intragenic regions (*Figure 3B*). Comparing the binding profiles of BRD9 and SS18-SSX1 demonstrated that these proteins co-localize extensively on the synovial sarcoma genome. Indeed, a clear majority of all identified BRD9 and SS18-SSX1 binding sites overlap (*Figure 3C*), and there is a tight correlation in BRD9 and SS18-SSX1 occupancy genome-wide (*Figure 3—figure supplement 1B*). Additional ChIP-seq analyses of RNA polymerase II (RNAPII) and the histone modification H3K27Ac further demonstrates that BRD9 and SS18-SSX1 bind virtually all active gene promoters and enhancer elements (*Figure 3D and E*). With little evidence of significant binding at inactive genomic loci. Two recent studies characterised gene expression signatures defining synovial sarcoma tumours, demonstrating that the SS18-SSX fusion directly binds many of these genes (*McBride et al., 2018*; *Banito et al., 2018*). Importantly, we found a significant overlap between these previous SS18-SSX1 ChIP studies and our own epitope tag knock-in mediated SS18-SSX1 and BRD9 ChIP-seq experiments (*Figure 3F* and *Figure 3—figure supplement 1C*). Considering that BRD9 may be present in only ~15% of SS18-SSX containing complexes, such broad co-localisation is remarkable; and suggests BRD9 containing complexes play an important role in supporting SS18-SSX function genome-wide.

## BRD9 bromodomain inhibition

Several recent studies have described the development of potent small-molecule inhibitors of the BRD9 bromodomain (*Hohmann et al., 2016*; *Martin et al., 2016*; *Theodoulou et al., 2016*). To test the feasibility of small-molecule mediated targeting of the BRD9 bromodomain as a therapeutic

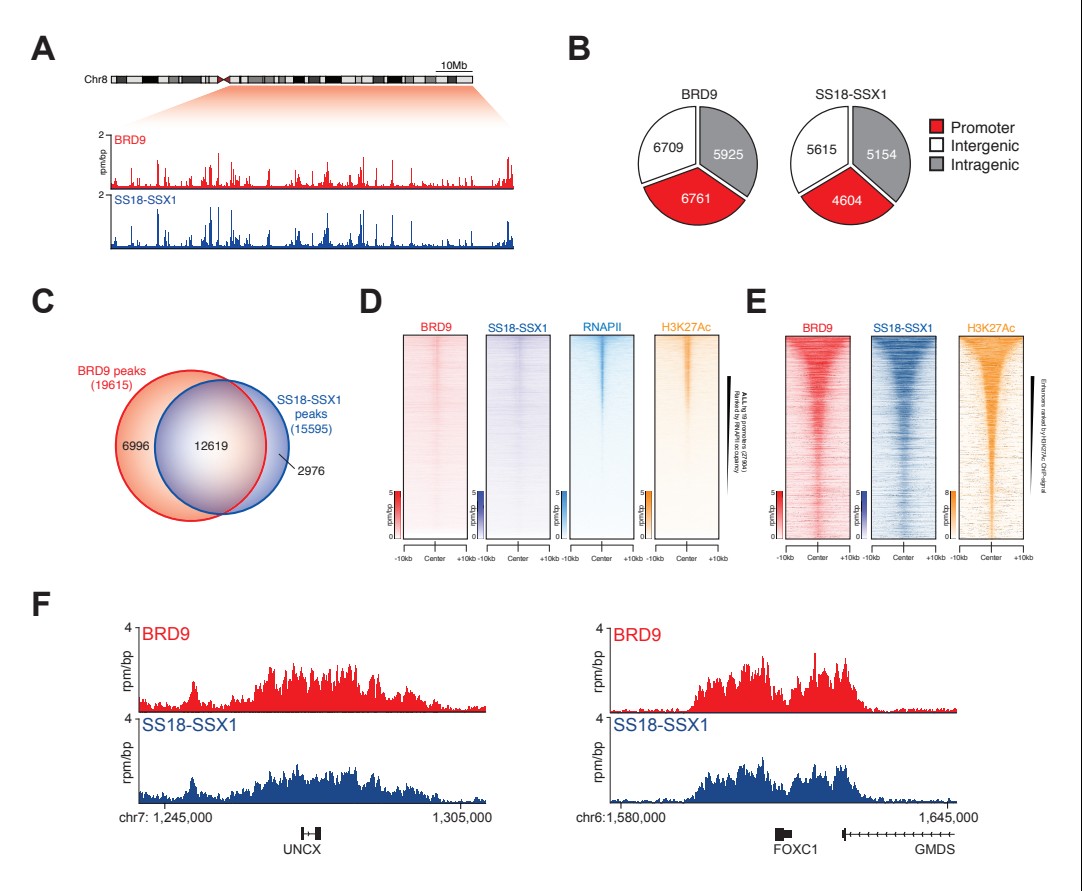

**Figure 3.** SS18-SSX1 and BRD9 co-bind the synovial sarcoma genome. (**A**) Genomic tracks showing BRD9 and SS18-SSX1 ChIP-seq signal on the 98 Mb right arm of chromosome eight in HSSYII cells. The chromosome eight ideogram is displayed above the gene tracks with the relevant region highlighted in red. (**B**) Pie charts representing the distribution of BRD9 and SS18-SSX1 binding sites on the synovial sarcoma genome. (**C**) Venn diagram overlaps of all identified BRD9 and SS18-SSX1 ChIP-seq peaks in HSSYII cells. (**D**) Tornado plots showing BRD9, SS18-SSX1, RNAPII and H3K27Ac ChIP-signal ±10 kb of all hg19 gene promoters in HSSYII cells. Promoters are ranked by RNAPII ChIP signal. (**E**) Tornado plots showing BRD9, SS18-SSX1 and H3K27Ac ChIP-signal ±10 kb of all active enhancers (defined by H327Ac) in HSSYII cells. (**F**) Tracks showing BRD9 and SS18-SSX1 ChIP-seq occupancy at the indicated genomic loci in HSSYII cells.

DOI: https://doi.org/10.7554/eLife.41305.018

The following figure supplement is available for figure 3:

**Figure supplement 1.** BRD9 and SS18-SSX1 co-localise genome-wide.

DOI: https://doi.org/10.7554/eLife.41305.019

approach in synovial sarcoma we performed dose response experiments using two independent BRD9 inhibitors, BI7273 and I-BRD9. Consistent with our genetic data, synovial sarcoma cells were more sensitive to BRD9 bromodomain inhibition compared to other pediatric sarcomas (***Figure 4A***). However, these effects were modest with growth IC50 values in the µM range. Interestingly, spike-in normalized BRD9 ChIP-seq (ChIP-Rx) performed in BI7273 treated cells demonstrated that while the chromatin occupancy of BRD9 is reduced in inhibitor treated cells some BRD9 remains associated with chromatin (***Figure 4B***). This indicates that BRD9 does not rely exclusively on bromodomain function to associate with chromatin. Consistent with this, mutational inactivation of the BRD9 bromodomain also leads to an incomplete loss of BRD9 binding across the genome (***Figure 4—figure supplement 1A and B***). Significantly, the ability of BRD9 to incorporate into the BAF complex is required for chromatin association, since deleting the BAF complex interaction domain (aa311-345) leads to a similar reduction in chromatin binding as bromodomain deletion (***Figure 4C*** and ***Figure 4—source data 1***). Taken together, these data indicate that BRD9, as part of the BAF complex,

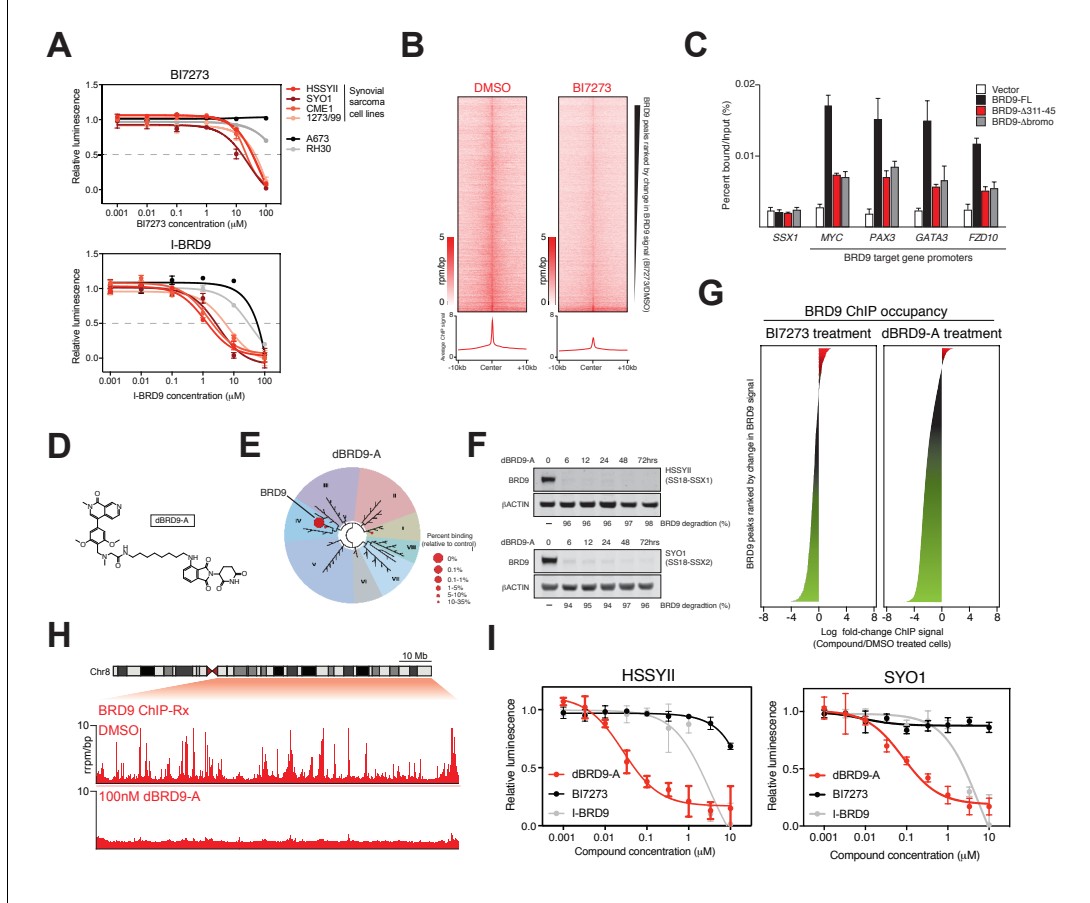

**Figure 4.** Complete ablation of BRD9 function by targeted protein degradation. (**A**) Cellular viability dose-response data in the indicated panel of cell lines treated with the BRD9 bromodomain inhibitors BI7273 (top) or I-BRD9 (bottom). Mean ± s.d., n = 3. (**B**) Tornado plots and meta-tracks representing BRD9 ChIP-Rx signal in control (DMSO) and BI7273 treated (24 hr) HSSYII cells. (**C**) ChIP-qPCR analysis of 3xHA epitope tagged full-length BRD9, Δ311–345 BRD9 or Δ bromodomain BRD9 at the indicated gene promoters in HSSYII cells. Mean ± s.d., n = 3. (**D**) Chemical structure of our BRD9 degrader compound dBRD9-A. (**E**) Selectivity of phage-displayed bromodomain displacement by dBRD9-A (Bromoscan). (**F**) Western blot analysis of the indicated proteins, in two independent synovial sarcoma cell lines following treatment with dBRD9-A at 100 nM for 6–72 hr. (**G**) Waterfall plot representing changes in BRD9 occupancy at BRD9 peak regions in ChIP-Rx experiments of BI7273 (10 μM) (left panel) or dBRD9-A (100 nM) (right panel) treated HSSYII cells following 24 hr treatment. (**H**) Tracks showing BRD9 ChIP-seq occupancy on the 98 Mb right arm of chromosome eight after DMSO or 100 nM dBRD9-A treatment. The chromosome eight ideogram is displayed above the gene tracks with the relevant region highlighted in red. (**I**) Cellular viability dose-response data in HSSYII and SYO1 cells treated with dBRD9-A or the BRD9 bromodomain inhibitors BI7273 or I-BRD9. Mean ± s. d., n = 3.

DOI: https://doi.org/10.7554/eLife.41305.020

The following source data and figure supplements are available for figure 4:

**Source data 1.** ChIP-qPCR data of HA-tagged BRD9 proteins - BRD9-FL, BRD9 Δbromo and BRD9 Δ311–345 - expressed in HSSYII cells.
DOI: https://doi.org/10.7554/eLife.41305.023
**Figure supplement 1.** Transcriptional regulation by BRD9 in SS cells.
DOI: https://doi.org/10.7554/eLife.41305.021
**Figure supplement 1—source data 1.** Fold-change of individual BAF complex members identified in SS18-SSX1 purifications from HSSYII cells treated with DMSO or dBRD9-A at 100 nM for 24 hr.
DOI: https://doi.org/10.7554/eLife.41305.022

can access chromatin in a bromodomain independent manner. Highlighting that bromodomain inhibition, while at least partially effective at blocking synovial sarcoma cell growth/survival, is unlikely to completely ablate the functional contributions of BRD9.

## Targeted degradation of the BRD9 protein

To completely inactivate BRD9 function we leveraged our recent success developing a targeted chemical degrader of BRD9 (*Remillard et al., 2017*). We created an optimized chemical analogue of our previous BRD9 degrader, dBRD9-A (*Figure 4D*). This molecule contains a more lipophilic alkyl linker and exhibits improved BRD9 degradation properties (data not shown). Importantly, dBRD9-A is a highly specific binder of the BRD9 bromodomain (*Figure 4E*); and elicits near complete BRD9 degradation at low nanomolar concentrations (*Figure 4F*). These degradation effects are dependent on the E3 ubiquitin ligase component CRBN, as well as BRD9 bromodomain engagement (*Figure 4—figure supplement 1B–C*). ChIP-Rx experiments demonstrate a far more robust loss of BRD9 binding across the genome following dBRD9-A treatment; compared to BI7273 treatment (*Figure 4G*). Indeed, essentially no BRD9 remains bound on chromatin following 24 hr of dBRD9-A treatment (*Figure 4H*). Significantly, BRD9 degradation leads to a greater therapeutic response than bromodomain inhibition (*Figure 4I*); consistent with the notion that BRD9 also functions independently of its bromodomain. Moreover, consistent with our genetic data other pediatric sarcoma subtypes are unaffected by BRD9 degradation (*Figure 4—figure supplement 1E–F*). Interestingly, the observed increase in therapeutic response in dBRD9-A treated cells is not due to destabilisation of the SS18-SSX fusion itself (*Figure 4—figure supplement 1G*). However, quantitative interactions proteomics of SS18-SSX containing complexes in dBRD9-A treated cells demonstrate that the GBAF members (GLTSCR1/L), in addition to BRD9, are lost from fusion protein containing complexes following BRD9 degradation (*Figure 4—figure supplement 1H*). This suggests that BRD9 is essential for the proper assembly of GBAF complexes; and that BRD9 degradation specifically disrupts this subclass of SS18-SSX containing complexes. Taken together, these data demonstrate that targeting BRD9 function with chemical degraders, rather than bromodomain inhibitors, is a more efficacious therapeutic approach in synovial sarcoma.

## BRD9 supports oncogenic transcription in synovial sarcoma

Synovial sarcoma cells treated with dBRD9-A undergo a progressive cell cycle arrest (*Figure 5A* and *Figure 5—source data 1*), which is further associated with an increase in Annexin–V positivity (*Figure 5B* and *Figure 5—source data 3*). Consistent with the on-target activity of dBRD9-A, swapping the BRD9 bromodomain for the closely related BRD7 bromodomain (63%, sequence identity) renders BRD9 and synovial sarcoma cells insensitive to dBRD9-A treatment (*Figure 5—figure supplement 1A–B* and *Figure 5—figure supplement 1—source data 1*). Using an in vivo synovial sarcoma xenograft model, we found that treatment of mice with dBRD9-A over 24 days inhibited tumour progression (*Figure 5C*). We confirmed in vivo pharmacodynamic activity of dBRD9-A in this system by immunoblotting BRD9 in tumour tissue derived from vehicle and dBRD9-A treated mice (*Figure 5D*). Mice treated with dBRD9-A did not suffer any overt side effects associated with treatment, retaining a normal body weight and blood counts (*Figure 5—figure supplement 1C–D* and and *Figure 5—figure supplement 1—source datas 2–3*). Next we performed cell count normalized RNA-seq analysis to understand why synovial cells are effected by BRD9 degradation. We performed these experiments 6 hr after dBRD9-A treatment (the earliest time point where we observe complete BRD9 degradation) to allow characterisation of the primary molecular changes following BRD9 loss. Strikingly, degradation of BRD9 primarily leads to down regulated gene expression (*Figure 5E* and *Figure 5—source data 3*). Using our H3K27Ac ChIP-seq data we identified the subset of genes associated with super enhancer (SE) elements; since studies have demonstrated that SE-associated genes are highly sensitive to transcriptional perturbation. Moreover, SEs drive expression of genes required for maintaining tumour cell identity (*Hnisz et al., 2013*; *Lovén et al., 2013*). Consistent with this, several genes associated with SEs in HSSYII cells including TWIST1 (*38*) and TLE1 (*22*) are known to play key functional roles in synovial sarcoma (*Figure 5F*). Moreover, expression of many of these genes has previously been linked to primary synovial sarcoma tumour phenotypes, defining both clinical and biological characteristics (*McBride et al., 2018*; *Banito et al., 2018*; *Francis et al., 2007*; *Sarver et al., 2015*; *Terry et al., 2007*; *Takahashi et al., 2014*; *Baird et al., 2005*). SEs have higher BRD9 and SS18-SSX1 occupancy levels compared to typical enhancers (*Figure 5—figure supplement 1E*); and BRD9 degradation leads to a preferential downregulation of SE associated gene expression (*Figure 5G*). Significantly, these genes depend on SS18-SSX1 to maintain their expression, since shRNA mediated knockdown of SS18-SSX1 leads to a collapse of SE associated gene

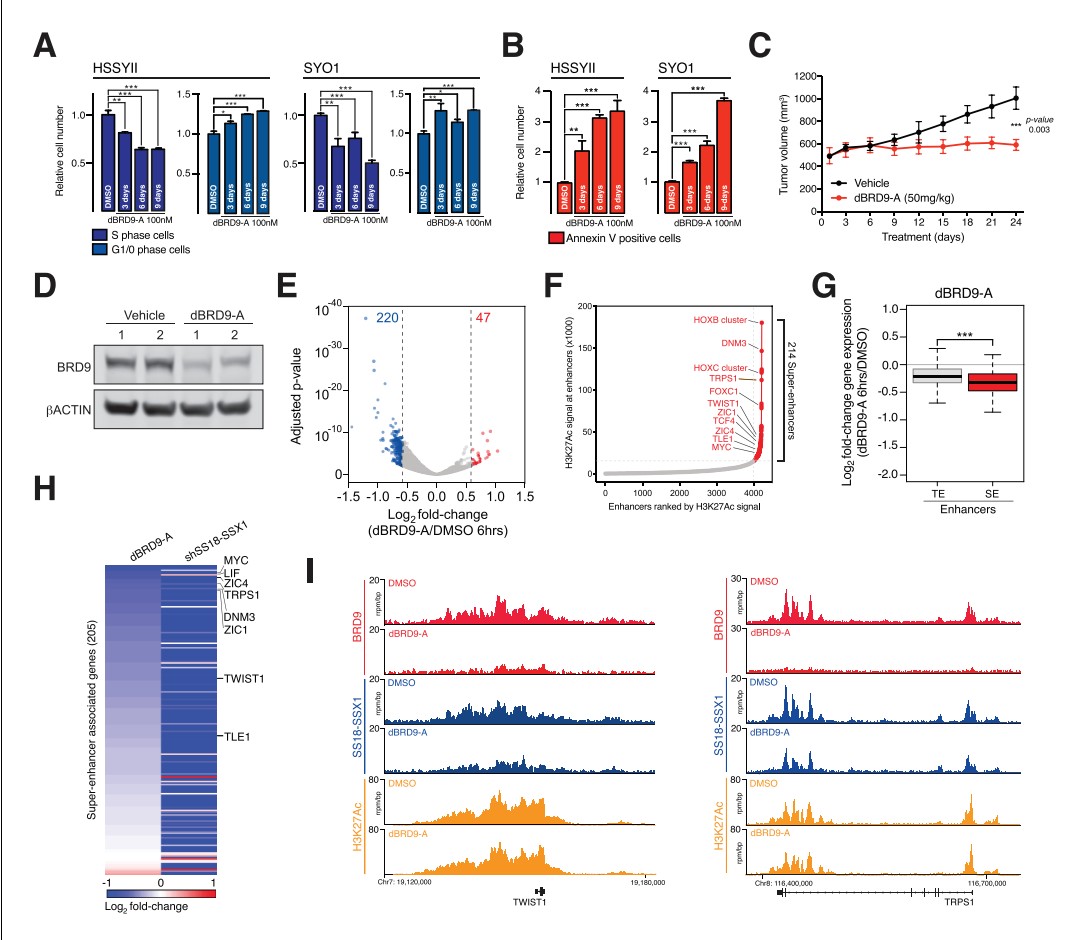

**Figure 5.** BRD9 degradation blocks synovial sarcoma tumour progression and oncogenic transcription. (**A**) Relative changes in cell cycle dynamics in two independent SS cell lines treated with dBRD9-A for 3/6/9 days at 100 nM. Mean ±s.d., n = 3 (**B**) Relative changes in Annexin-V positive cells in two independent SS cell lines treated with dBRD9-A for 3/6/9 days at 100 nM. Mean ± s.d., n = 3 (**C**) Tumour progression in a subcutaneous xenograft model of SS in control vehicle treated mice, and mice treated at 50 mg/kg dBRD9-A once daily for 24 days. Mean ± SEM, five mice per treatment group. P value is from 2way ANOVA (**D**) Western blot analysis of the indicated proteins in protein lysates derived from tumour tissue from two independent mice per treatment group as in panel c. (**E**) Volcano plot representing gene expression changes in HSSYII cells treated with dBRD9-A at 100 nM for 6 hr. The number of genes, the expression of which changes >1.5 fold up or down are indicated. (**F**) H3K27Ac ChIP-seq signal (rpm/bp) at all enhancer regions in HSSYII cell. Enhancers are ranked by increasing H3K27Ac signal. (**G**) Box plot representations of changes in gene expression amongst genes associated with typical enhancers and genes associated with super enhancers. P values are from Welch's two-tailed t-tests. ***p≤0.001. (**H**) Heat map representing changes in gene expression amongst all super enhancer associated genes in HSSYII cells following 6 hr dBRD9-A treatment at 100 nM, or HSSYII cells following infection with two independent SS18-SSX1 shRNAs for 96 hr. (**I**) Tracks showing BRD9 and SS18-SSX1 ChIP-seq occupancy at the indicated genomic loci in DMSO and dBRD9-A treated cells. Also shown is H3K27Ac ChIP-seq signal in untreated cells.

DOI: https://doi.org/10.7554/eLife.41305.024

The following source data and figure supplements are available for figure 5:

**Source data 1.** Induction of apoptosis in HSSYII and SYO1 cells treated with dBRD9-A at 100 nM over 9 days.
DOI: https://doi.org/10.7554/eLife.41305.029
**Source data 2.** Cell cycle dynamics of HSSYII and SYO1 cells treated with dBRD9-A at 100 nM over 9 days.
DOI: https://doi.org/10.7554/eLife.41305.030
**Source data 3.** Gene expression changes in HSSYII cells treated with dBRD9-A at 100 nM for 6 hr.
DOI: https://doi.org/10.7554/eLife.41305.031
**Figure supplement 1.** Targeted degradation of BRD9.
DOI: https://doi.org/10.7554/eLife.41305.025
**Figure supplement 1—source data 1.** Cell counts in dBRD9-A treatment experiments in HSSYII cells infected with an empty vector, a WT BRD9 expressing vector or a BRD9 bromodomain swap (BRD7 bromodomain) vector.
DOI: https://doi.org/10.7554/eLife.41305.026
**Figure supplement 1—source data 2.** Mouse weight measurement derived from mice treated with control (vehicle) of dBRD9-A at 50 mg/kg.

*Figure 5 continued on next page*

*Figure 5 continued*

DOI: https://doi.org/10.7554/eLife.41305.027

**Figure supplement 1—source data 3.** Presented are blood counts derived from DMSO and dBRD9-A treated mice 1 day prior to cessation of treatment.

DOI: https://doi.org/10.7554/eLife.41305.028

expression (*Figure 5H*). dBRD9-A treatment and consequential downregulation of transcription is further associated with reductions in SS18-SSX1 binding at SEs (*Figure 5I* and *Figure 5—figure supplement 1F*). Interestingly, since SS18-SSX1 and BRD9 directly bind the majority of active genes, these transcriptional perturbations amount to a relatively narrow impingement on the broader cohort of SS18-SSX downstream target genes (*Figure 5—figure supplement 1G*). However, these data demonstrate that BRD9 is required to maintain the SE associated oncogenic transcriptional program driven by SS18-SSX and the phenotypic impact of BRD9 degradation underlines the functional importance of this gene cohort. Most importantly, these data show that targeting BRD9 using our novel degrader compound directly perturbs underlying oncogenic mechanisms in this disease.

## Discussion

We've shown that BRD9 is an essential SS18-SSX fusion protein co-factor in synovial sarcoma. Our data indicate that through assembly into SS18-SSX containing complexes BRD9 supports oncogenic gene expression programs necessary for synovial sarcoma oncogenesis. This is likely achieved, at least in part, via bromodomain mediated interactions with chromatin regions marked with acetyl-lysine modifications. Indeed, we observe the greatest amount of BRD9 and SS18-SSX binding at super enhancer elements associated with high levels of H3K27Ac. BRD9 may promote and/or stabilize binding of SS18-SSX containing complexes at acetylated chromatin regions; since loss of BRD9 can lead to reduced fusion protein occupancy at some super enhancers. This induces perturbations of oncogenic gene expression programs driven by the fusion protein and robust therapeutic effects. This work highlights the first actionable therapeutic vulnerability, directly linked to SS18-SSX, in synovial sarcoma. These findings provide a rationale for future clinical investigations of BRD9 as a therapeutic target in synovial sarcoma patients.

This work demonstrates that direct targeting of oncogenic, SS18-SSX containing BAF complexes is a viable therapeutic approach in synovial sarcoma. Recent work has indicated that BRD9 assembles into a previously unreported form of the BAF complex, termed GBAF (*Alpsoy and Dykhuizen, 2018*). This complex is defined by the presence of either GLTSCR1 or GLTSCR1L and BRD9; and also lacks SNF5 and an ARID component. This altered complex assembly suggests potential functional specialization within BRD9 containing complexes. Therefore, it will be important to perform detailed structure/function studies to better understand the molecular contributions of these novel complexes. Interestingly, our finding that GLTSCR1 and GLTSCR1L are lost from SS18-SSX containing complexes following BRD9 degradation already suggests that BRD9 is essential for assembly of GBAF complexes. The specificity of the observed dependency on BRD9 containing complexes in synovial sarcoma cells is quite remarkable and bodes well for potential clinical applications in patients. Interestingly, a recent report indicated that malignant rhabdoid tumour (MRT) cells are also dependent on BRD9 function (*Krämer et al., 2017*). Interestingly, MRTs and synovial sarcoma share a common feature, that being loss of SNF5. In essentially 100% of MRTs *SNF5* is subject to biallelic inactivation; while in synovial sarcoma assembly of SS18-SSX into BAF complexes leads to eviction and proteasomal degradation of SNF5. Since SNF5 is absent in BRD9 containing complexes it is tempting to speculate that loss of SNF5 (by genetic or biochemical means) shifts the balance of BAF complex assembly to a more GBAF-like state. This could explain the shared dependency on BRD9 function in these malignancies. Several additional cancers, including bladder cancer and uterine corpus endometrial carcinoma, have a high frequency of mutations effecting the genes encoding ARID1A/B (*Bailey et al., 2018*). Loss of function mutations in these components could potentially shift the dynamics of BAF complex assembly toward a GBAF-like state. Therefore, it will be important to test the efficacy of BRD9 targeting in other cancers with BAF complex mutations.

Most cancer treatments target processes important in normal and cancer cells, therefore toxicities resulting in debilitating side-effects remains problematic. Fusion gene driven cancers present a

relatively unique opportunity to target cancer cell specific processes since oncogenic fusion proteins are present only in malignant cells. Understanding mechanisms related to fusion protein function may provide opportunities to develop therapies targeting underlying pathologies with limited effects on normal tissues. Our work demonstrates the importance of BRD9 in supporting SS18-SSX function and oncogenic gene expression in synovial sarcoma cells. Currently synovial sarcoma has few effective treatment options, and advanced forms of the disease have very poor overall survival. This study provides a rationale for development of BRD9 degradation as a novel therapeutic approach and potentially assessment in patients suffering with this disease. We demonstrate that degradation of BRD9, a member of an oncogenic multi-protein complex in synovial sarcoma, has a more profound effect on cancer cell survival than small-molecule mediated inhibition. This is an important point since inhibition of chromatin regulators such as EZH2, DOT1L and LSD1, which also exist in stable multi-protein complexes is currently under clinical investigation in several cancers (*Brien et al., 2016*; *Cai et al., 2015*). Our findings suggest that inhibition, while effectively blocking a single functionality within a target protein, may provide a relatively ineffective means to block protein complex function as a whole. Scaffolding/other non-inhibited functions of a target protein may remain unaffected, allowing the target to continue supporting complex function. Therefore, degradation of proteins within multi-protein complexes may be a more efficacious approach in many cases. Importantly, the potent and selective small-molecule inhibitors that already exist for proteins such as EZH2 and DOT1L will provide a basis for the development of novel protein degraders targeting these proteins.

## Materials and methods

**Key resources table**

| Reagent type (species) or resource | Designation | Source or reference | Identifiers | Additional information |
|---|---|---|---|---|
| Cell line (*Homo sapiens*) | HEK293T | ATCC | RRID:CVCL_0063 | |
| Cell line (*Homo sapiens*) | HSSYII | | | Provided from the laboratory of Stefan Frohling |
| Cell line (*Homo sapiens*) | SYO1 | | | Provided from the laboratory of Stefan Frohling |
| Cell line (*Homo sapiens*) | 1273/99 | | | Provided from the laboratory of Stefan Frohling |
| Cell line (*Homo sapiens*) | A673 | ATCC | RRID:CVCL_0080 | |
| Cell line (*Homo sapiens*) | CME1 | | | Provided from the laboratory of Stefan Frohling |
| Cell line (*Homo sapiens*) | SKNMC | ATCC | RRID:CVCL_0530 | |
| Cell line (*Homo sapiens*) | RH30 | ATCC | RRID:CVCL_0041 | |
| Cell line (*Homo sapiens*) | RH41 | DSMZ | RRID:CVCL_2176 | |
| Antibody | BRD9, rabbit polyclonal | Bethyl Laboratories | RRID:AB_11218396 | Western blotting (1:2500) and IP (5μgs) |
| Antibody | HA, rabbit monoclonal | Cell Signalling Technologies | RRID:AB_1549585 | Western blotting (1:1000) and ChIP (5-10μgs) |
| Antibody | ACTIN, mouse monoclonal | Cell Signalling Technologies | RRID:AB_2750839 | Western blotting (1:5000) |

*Continued on next page*

*Continued*

| Reagent type (species) or resource | Designation | Source or reference | Identifiers | Additional information |
|---|---|---|---|---|
| Antibody | CRBN, rabbit polyclonal | Proteintech | RRID:AB_2085739 | Western blotting (1:1000) |
| Antibody | V5, rabbit polyclonal | Bethyl Laboratories | RRID:AB_67586 | Western blotting (1:1000) |
| Antibody | SSX1, rabbit polyclonal | MyBioscience | RRID:AB_2750841 | IP (10µgs) |
| Antibody | SSX2, rabbit polyclonla | MyBioscience | RRID:AB_2750840 | IP (10µgs) |
| Antibody | SS18, rabbit polyclonal | Santa Cruz Biotechnology | RRID:AB_2195154 | Western blotting (1:500) |
| Antibody | H3K27Ac, rabbit polyclonal | Abcam | RRID:AB_2118291 | ChIP (5µgs) |
| Antibody | RNAPII, mouse monoclonal | Diagenode | RRID:AB_2750842 | ChIP (10µs) |
| Other | HA | Pierce | RRID:AB_2749815 | IP affinity resin |
| Other | V5 | Sigma Aldrich | RRID:AB_10062721 | IP affinity resin |
| Chemical compound, drug | dBRD9-A | This study | | |
| Chemical compound, drug | BI7273 | Cayman Chemical | 20311 | |
| Chemical compound, drug | I-BRD9 | Cayman Chemical | 17749 | |
| Chemical compound, drug | X-termeGENE 9 | Sigma Aldrich | 6365809001 | |
| Chemical compound, drug | Formaldehyde | Fisher Scientific | BP531-500 | |
| Chemical compound, drug | DSG | Pierce | 20593 | |
| Chemical compound, drug | ATPLite 1-Step | Perkin Elmer | 6016731 | |
| Recombinant DNA reagent | pPAX2 | Addgene | 12260 | |
| Recombinant DNA reagent | pCMV-VSV-G | Addgene | 8454 | |
| Recombinant DNA reagent | pLEX305 | Addgene | 41390 | |
| Recombinant DNA reagent | pLEX305-3xHA | This study | | |
| Recombinant DNA reagent | LRG2.0T | This study | | Provided form the laboratory of Chris Vakoc |
| Recombinant DNA reagent | SGEN | MSKCC RNAi core facility | | |
| Recombinant DNA reagent | pCR8 | Invitrogen | K250020 | |
| Recombinant DNA reagent | pCR8-BRD9 (and derivatives) | This study | | |
| Commercial assay or kit | 4–12% Bis-Tris gels | Invitrogen | NW04127BOX | |
| Commercial assay or kit | Q5 Site-Directed mutagenesis kit | NEB | E0554S | |

*Continued on next page*

*Continued*

| Reagent type (species) or resource | Designation | Source or reference | Identifiers | Additional information |
|---|---|---|---|---|
| Commercial assay or kit | ThruPlex DNA-seq kit | Rubicon Genomics | R400427 | |
| Commercial assay or kit | Tapestation D1000 screentape | Agilent | 5067–5584 | |
| Commercial assay or kit | NextSeq 500 High Output v2 | Illumina | FC-404–2005 | |
| Commercial assay or kit | RNeasy mini-kit | Qiagen | 74106 | |
| Commercial assay or kit | ERCC spike-in controls | Ambion | 4456740 | |
| Commercial assay or kit | NEBNext Ultra RNA library prep kit | NEB | E7530L | |
| Commercial assay or kit | BD Pharmingen BrdU Flow kit | BD | 559619 | |
| Commercial assay or kit | BD Annexin V Apoptosis detection kit | BD | 556547 | |
| Software, algorithm | ChIP and RNA-seq analysis | Basepair | | www.basepair.io |
| Software, algorithm | ChIP-seq data visualisation | EaSeq | | https://easeq.net |
| Strain, strain background (*Mus musculus*) | BALB/c (Foxn1nu) | Charles River Laboratory | CAnN. Cg-*Foxn1nu*/Crl | |

## Cell culture and lentiviral production

All cell lines were maintained at 37°C in a humidified incubator. Lentiviral packaging HEK293T, synovial sarcoma (HSSYII, SYO1 and 1273/99) and Ewing's sarcoma (A673) cell lines were cultured in DMEM (Gibco) media supplemented with 10% heat-inactivated fetal bovine serum (FBS), 1% Penicillin-Streptomycin and 12.5 ug/ml Plasmocin. Synovial sarcoma (CME1), Ewing's sarcoma (SKNMC) and rhabdomyosarcoma (RH30 and RH41) cells were cultured in RPMI (Gibco) media supplemented with 10% heat-inactivated fetal bovine serum (FBS), 1% Penicillin-Streptomycin and 12.5 ug/ml Plasmocin. Cell lines were tested regularly for mycoplasma contamination and tested negative in all cases. Lentiviral supernatants were generated by co-transfection of HEK293T cells with a lentiviral expression vector (cDNA, sgRNA or shRNA) with viral packaging (PAX2) and envelope (VSV-G) vectors using the X-tremegene transfection reagent (Roche) in accordance with the manufacturer's instructions. Viral supernatants were collected between 24–48 hr post-transfection and used directly for infection of target cells after filtering through a 0.45 µm syringe filter and addition of 8.5 µg/ml Polybrene.

## Pooled CRISPR screening and data analysis

The human epigenetic domain U6-sgRNA-EFS-GFP targeting library was pooled at equimolar ratio and used to generate lentiviral supernatant as described above. A dilution series of this virus correlated with GFP positivity in infected cells, and was used to derive an accurate viral multiplicity of infection (MOI). The total number of synovial and Ewing's sarcoma target cells for infection was chosen to achieve at least 500-fold representation of each sgRNA in the initially infected cell population. To ensure that a single sgRNA was transduced per cell, the viral volume for infection was chosen to achieve an MOI of 0.3–0.4. Genomic DNA was extracted at the indicated time points using QiAamp DNA mini kit (Qiagen #51304), following the manufacturer's instructions. To maintain >500 × sgRNA library representation, 16–20 independent PCR reactions were used to amplify the sgRNA cassette, which were amplified for 20 cycles with 100–200 ng of starting gDNA using the 2 × Phusion Master Mix (Thermo Scientific #F-548). The PCR products were pooled and end repaired with T4 DNA polymerase (NEB), DNA polymerase I (NEB), and T4 polynucleotide kinase (NEB). An A overhang was

added to the end-repaired DNA using Klenow DNA Pol Exo- (NEB). The DNA fragment was then ligated with diversity-increased barcoded Illumina adaptors followed by five pre-capture PCR cycles. Barcoded libraries were pooled at equal molar ratio and subjected to massively parallel sequencing using a Mi-Seq instrument (Illumina) using paired-end 150 bp reads (MiSeq Reagent Kit v2; Illumina MS-102–2002). The sequence data were trimmed to contain only the sgRNA sequence then mapped to the reference sgRNA library without allowing any mismatches. The read counts were then calculated for each individual sgRNA. To compare the differential representation of individual sgRNAs between day 3 and day 15 time points, the read counts for each sgRNA were normalized to the counts of the negative control ROSA26 sgRNA.

## Cloning and mutagenesis

The human full-length BRD9 cDNA was PCR amplified from MGC clone 5428011 and inserted in the Gateway cloning compatible entry vector pCR8/GW/TOPO (Invitrogen, K250020) in accordance with the manufacturer's instructions. Clone integrity was confirmed by sanger sequence. Mutagenesis of the wildtype BRD9 sequence was performed using pCR8-BRD9 as template and the Q5 Site-Directed Mutagenesis Kit (NEB, E0554S) in accordance with the manufacturer's instructions. Sequence verified BRD9 ORF sequences were subsequently cloned into the Gateway expression vector pLEX305 (Addgene vector, 41390) which had been engineered to contain an N-terminal 3xHA epitope tag using LR clonase (Invitrogen, 12538120).

## Immunoprecipitation

Immunoprecipitations were performed as previously described (*Brien et al., 2015*). Briefly, nuclear pellets were lysed in buffer C containing protease inhibitors (20 mM HEPES at pH 7.6, 20% [v/v] glycerol, 0.42 M NaCl, 1.5 mM $MgCl_2$, 0.2 mM EDTA, aprotinin 1 µg $mL^{-1}$, leupeptin 10 µg $mL^{-1}$, PMSF 1 mM) and subsequently dialyzed against buffer C-100 (20 mM HEPES at pH 7.6, 20% [v/v] glycerol, 0.2 mM EDTA, 100 mM KCl, 1.5 mM MgCl2, 0.2 mM EDTA). Antibody-coupled beads were incubated with dialyzed nuclear extracts containing 250 U of benzonase (Sigma) for 3 hr at 4°C. Beads were then washed, and elutions were performed with 1xLDS buffer, 1 mg/mL HA peptide or 1 mg/mL V5 peptide (Sigma).

## Mass spectrometry

In-solution tryptic digestions were performed as described previously(*Wiśniewski et al., 2009*). Peptides were analysed with a Q-Exactive mass spectrometer coupled with an EASY-nLC HPLC system (Thermo Fisher) and an in-house packed C18 column (New Objective). Parent ion spectra (MS1) were measured at resolution 70,000, AGC target 3e6. Tandem mass spectra (MS2, up to 10 scans per duty cycle) were obtained at resolution 17,500, AGC target 5e4, collision energy of 25. All mass spectrometry data were processed using the MaxQuant software, version 1.3.0.5 (*49*). The following search parameters were used; Fixed Mod: carbamidomethylation, Variable Mods: methionine oxidation, Trypsin/P digest enzyme, Precursor mass tolerances six ppm, Fragment ion mass tolerances 20 ppm, Peptide FDR 1%, Protein FDR 1%.

## Quantitative interaction proteomics in dBRD9-A treated cells

### On bead digestion and mass spectrometry

After the pulldown, the beads were resuspended in elution buffer (2M Urea, 100 mM Tris pH 8, 10 mM DTT) and incubated 20 min on a shaker (1300 rpm) at RT. After incubation, iodoacetamide was added to a final concentration of 50 mM, followed by 10 min shaking in the dark at RT. Partial digestion and elution from the beads was initiated by adding 0.25 µg Trypsin (Promega; V5113) for 2 hr. The supernatant containing the IP samples was collected and the beads were resuspended in 50 µl elution buffer followed by a 5 min incubation shaking at RT. Both supernatants were combined and 0.1 µg Trypsin was added followed by overnight incubation at RT. The digestion was stopped by adding TFA (final concentration 0.5%). The resulting digested samples were desalted and purified using StageTips (*Rappsilber et al., 2007*). The peptides were eluted from StageTips with buffer B (80% acetonitrile, 0.1% formic acid), concentrated to 5 µL by SpeedVac centrifugation at room temperature, and filled up to 12 µL using buffer A (0.1% formic acid). Pulldown samples were measured using a gradient from 9–32% Buffer B for 114 min followed by washes at 50% then 95% Buffer B,

resulting in total 140 min data collection time. Mass spectra were recorded on an LTQ-Orbitrap Fusion Tribrid mass spectrometer (Thermo Fisher Scientific). Scans were collected in data-dependent top speed mode with dynamic exclusion set at 60 s.

## Mass spectrometry analysis

Thermo RAW files were analyzed with MaxQuant version 1.5.1.0 using default parameters. Searches were performed against the Uniprot mouse proteome, downloaded at June 2017. Additional parameters that were enabled were match-between-runs, label-free quantification (LFQ) and IBAQ. After filtering for proteins that were present at least in all replicates of one condition, LFQ values were log2 transformed and missing values were imputed in Perseus using default parameters (width = 0.3, shift = 1.8). Statistical outliers for the pulldowns were determined using a two-tailed *t*-test. Multiple testing correction was performed using a permutation-based false discovery rate (FDR) method in Perseus. Volcano plots and stoichiometry calculations were performed as described previously (*Smits et al., 2013*).

## Chromatin immunoprecipitation

Cells for H3K27Ac and RNAPII ChIPs were fixed using 1% formaldehyde at room temperature for 10 mins. Formaldehyde crosslinking was quenched by adding Glycine to a final concentration of 0.125M directly to the fixation solution, followed by an additional 5 min incubation at room temperature. Cells for anti-HA (BRD9/SS18-SSX1) ChIPs were subjected to a 2-stage fixation; cells were initially fixed for 30 mins at room temperature using 0.5 mM DSG, followed by an additional 10 mins at room temperature using 1% formaldehyde. Formaldehyde crosslinking was quenched as outlined above. Fixed cells were washed 2X with ice-cold PBS and pelleted by centrifugation. Nuclei were extracted by resuspending fixed cell pellets in LB1 buffer (50 mm HEPES,140mm NaCl, 1 mm EDTA, 10% Glycerol, 0.5% NP40, 0.25% Triton X100) containing 1X protease inhibitor cocktail (Biotools, B14002), followed by 10 mins incubation. Cells were pelleted by centrifugation and resuspended in LB2 buffer (10 mM Tris ph8.0, 200 mM NaCl, 1 mM EDTA, 0.5 mM EGTA) containing 1X protease inhibitor cocktail. Extracted nuclei were lysed using Covaris shearing buffer (0.1% SDS, 1 mM EDTA and 10 mM Tris pH 8.0) containing 1X protease inhibitor cocktail. Nuclei were lysed at a concentration of 10–30 million cells/ml in shearing buffer and sonicated in a Covaris E220, 1 ml AFA milltubes (with fiber), Water level = 5, Duty Cycle = 5%, Peak Incidence Power = 140W, Cycle per burst = 200 for 16mins. Sonicated samples were pre-cleared by centrifugation at 14000 rpm for 15mins at 4°C. A 0.25X volume of 5X ChIP buffer (250 mM HEPES, 1.5 M NaCl, 5 mM EDTA pH 8.0, 5% Triton X-100, 0.5% DOC, and 0.5% SDS) was added to pre-cleared lysates, and these samples used directly for immunoprecipitations. For spike-in normalized ChIP experiments (ChIP-Rx) a 1:10 vol of fixed/sonicated chromatin derived for a mouse NIH3T3 cell line expressing a 3x HA epitope tagged BRD9 was added to each sample prior to the immunoprecipitation step.

## ChIP-seq library preparation and sequencing

ChIP purified DNA was quantified using a Qubit fluorimeter (Invitrogen), and 2–50 ng of DNA/ChIP was used to generate ChIP-seq libraries with the ThruPLEX DNA-seq kit (Rubicon Genomics, R400427). Library DNA was quantified using the Qubit, and size distributions were ascertained on a Tapestation (Agilent) using the D1000 ScreenTape assay reagents (Agilent, 5067–5583). This information was used to calculate pooling ratios for multiplex library sequencing. Pooled libraries were diluted and processed for 75 bp single-end sequencing on an Illumina NextSeq instrument using the NextSeq 500 High Output v2 kit (75 cycle) (Illumina, FC-404–2005) in accordance with the manufacturer's instructions.

## Cell count RNA-seq library prep and sequencing

Total RNA was isolated from cells using the RNeasy Mini Kit (Qiagen, 74106) in accordance with the manufacturer's instructions. ERCC spike-in controls were added to isolated RNA to facilitate cell count normalization of RNA-sequencing data. The quality of extracted RNA was confirmed using a Bioanalyzer (Agilent) and 1 µg of total RNA was used/sample as library prep input. Libraries were generated using the NEBNext Ultra RNA Library Prep kit for Illumina (NEB, E7530L) in accordance with the manufacturer's instructions. Library DNA was quantified using the Qubit, and size

distributions were ascertained on a Tapestation (Agilent) using the D1000 ScreenTape assay reagents (Agilent, 5067–5583). This information was used to calculate pooling ratios for multiplex library sequencing. Pooled libraries were diluted and processed for 75 bp single-end sequencing on an Illumina NextSeq instrument using the NextSeq 500 High Output v2 kit (75 cycle) (Illumina, FC-404–2005) in accordance with the manufacturer's instructions.

## ChIP and RNA-seq data analysis

ChIP-seq analysis was performed using pipelines on the omics analysis platform Basepair (http://www.basepair.io). ChIP fastq files were trimmed to remove adapter and low quality sequences using trim_galore and aligned to the UCSC genome assembly hg19 using Bowtie (version 2.1.0). For spike-in normalized ChIP-seq experiments reads were separately aligned to hg19 and mm9 using Bowtie. Duplicate reads were removed using Picard Mark Duplicates. Peaks were detected using MACS (version 1.4) using a p value cutoff was set to $10^{-5}$. Peaks were annotated to genomic features (Promoter, Gene body, Intergenic) using custom scripts on the Basepair platform, based on the UCSC database for hg19. ChIP-seq data visualisations were generated using the EaSeq analysis software (Lerdrup et al., 2016).

RNA-seq fastq files were aligned to NCBI37/hg19 and normalized using STAR. Differential expression data were obtained using the DEseq algorithm. These analyses were all done through the Basepair analysis platform (http://www.basepair.io).

## Immunoblotting

Whole cell protein samples were prepared in RIPA buffer (25 mM Tris-HCl. pH7.6, 150 mM NaCl, 1% NP-40, 1% Sodium Deoxycholate, 0.1% SDS) containing 1X protease inhibitor cocktails. Protein lysates were separated on pre-cast Bolt 4–12% Bis-Tris Plus Gels (Invitrogen, NW04127BOX) and transferred to nitrocellulose membranes. Membranes were subsequently probed using the relevant primary and secondary antibodies and relative protein levels were determined using the Odyssey CLx Imager (LI-COR).

## Cellular viability, cell cycle and apoptosis analysis

For dose response viability assays, cells were plated in 96-well tissue culture plates (1000 cells/well) in media containing DMSO or the desired concentration or each compound. Media was changed every 3 days up to a total of 9 days, at which point the ATPlite 1-Step luminescence assay system (PerkinElmer, 6016731) was used to determine ATP-dependent luminescence as an approximation of cellular viability. For cell cycle and apoptosis analysis cells were initially seeded on 10 cm dishes in media containing DMSO or 100 nM dBRD9-A and cultured/passaged in this media for a total of 9 days. For cell cycle analysis control and treated cells were harvested at 3/6/9 days and processed for FACs analysis using the BD Pharmingen BrdU Flow kit (BD, 559619) in accordance with the manufacturer's instructions. For apoptosis analysis cells were harvested at 3/6/9 days (using Accutase to maintain cell membrane integrity) and processed for FACs analysis using the BD Annexin V Apoptosis Detection kit (BD, 556547) in accordance with the manufacturer's instructions. Stained cells were analysed on a BD LSRFortessa Cell Analyzer and data processed using FlowJo software.

## Mouse experiments

4–6 week old female BALB/c (Foxn1[nu]) were purchased from Charles River Laboratories. For xenograft experiments mice were subcutaneously injected with 5 million synovial sarcoma cells in a 50/50 mix of culture media/matrigel. For treatment experiments dBRD9-A (50 mg/kg) was administered once daily via intraperitoneal injection, over a total of 24 days. All experiments described were approved by and adhered to the guidelines of the Dana Farber Cancer Institute animal care and use committee.

## Antibodies

### Antibodies used for western blotting

rabbit anti-BRD9 polyclonal, Bethyl Labs (catalogue number: A303-781A), rabbit anti-HA monoclonal, Cell Signaling Technology (catalogue number: 3724S), mouse anti-ACTIN monoclonal, Cell Signaling Technology (catalogue number: 3700S), rabbit anti-CRBN polyclonal, Proteintech (catalogue

number: 11435–1-AP), rabbit anti-V5 polyclonal, Bethyl Labs (catalogue number: A190-220A). Goat anti-Rabbit IgG polyclonal, LI-COR (catalogue number: 925–32211) and goat anti-Mouse IgG polyclonal, LI-COR (catalogue number: 926–68070).

## Antibodies used for IP
mouse anti-HA monoclonal magnetic beads, Pierce (catalogue number: 88837), mouse anti-V5 monoclonal agarose beads, Sigma (catalogue number: A7345-1ML), rabbit anti-BRD9 polyclonal, Bethyl Labs (catalogue number: A303-781A), rabbit anti-SSX1 polyclonal, MyBiosource (catalogue number: MBS9408371), rabbit anti-SSX2 polyclonal, MyBiosource (catalogue number: MBS9127222).

## Antibodies used for ChIP
rabbit anti-HA monoclonal, Cell Signaling Technology (catalogue number: 3724S), rabbit anti-H3K27Ac polyclonal, Abcam (catalogue number: ab4729), mouse anti-RNAPII monoclonal, Diagenode (catalogue number: C15100055-100).

## Chemical synthesis of dBRD9-A
### 4-bromo-2-methyl-2,7-naphthyridin-1(2H)-one
To a fine suspension of 4-bromo-2-methyl-2,7-naphthyridin-1(2H)-one (996 mg, 4.43 mmol, 1.0 eq) and Cesium Carbonate (4330 mg, 13.3 mmol, 3.0 eq) in THF (17.7 mL) was added Iodomethane (551 µL, 8.86 mmol, 2.0 eq) and stirred at RT. After 22 hr, the mixture was concentrated in vacuo, and the resulting residue dissolved in DCM. Insoluble material was filtered and washed with both DCM and water before being discarded. Organic filtrate was collected (approx. 150 mL), washed three times with deionized water (30 mL), and finally with saturated brine (30 millileters), before being dried over $Na_2SO_4$ and concentrated in vacuo to give the desired product as an off-white solid (1038 mg, 98%).

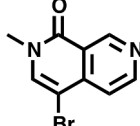

**Chemical structure 1.** 4-bromo-2-methyl-2,7-naphthyridin-1(2H)-one.
DOI: https://doi.org/10.7554/eLife.41305.032

**1H NMR** 1H NMR (500 MHz, DMSO-$d_6$) δ = 9.36 (s, 1H), 8.88 (s, 1H), 8.25 (s, 1H), 7.61 (s, 1H), 3.54 (s, 3H).
**Lcms**: 239 (M)

### tert-butyl 2-((4-bromo-2,6-dimethoxybenzyl)(methyl)amino)acetate
Sarcosyl tert-butyl ester hydrochloride (556 mg, 3.06 mmol, 1.5 eq) was dissolved in a solution of NaOAc (251 mg, 3.06 mmol, 1.5 eq), in DCM (8.2 mL), before 167 µL AcOH (2.04 mmol, 1.0 eq) was added, followed by 4-bromo-2,6-dimethoxybenzaldehyde (500 mg, 2.04 mmol, 1.0 eq). The mixture was stirred for 10 min before sodium triacetoxy borohydride was added in one portion (864.8 mg, 4.08 mmol, 2.0 eq), and the mixture stirred for 18 hr. The reaction was basified to approximately pH 11 with 1M $K_2CO_3$ and extracted 4 times with DCM (10 mL). The combined organics were washed with deionized water (10 mL), and saturated brine (10 mL), before being dried over $Na_2SO_4$ and concentrated in vacuo to give the desired product as an off-white solid (725 mg, 95%).

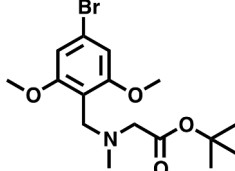

**Chemical structure 2** tert-butyl 2-((4-bromo-2,6-dimethoxybenzyl)(methyl)amino)acetate.
DOI: https://doi.org/10.7554/eLife.41305.033

**1H NMR** (500 MHz, Chloroform-d) δ = 6.69 (s, 2H), 3.81 (s, 2H), 3.79 (s, 6H), 3.21 (s, 2H), 2.41 (s, 3H), 1.48 (s, 9H).

**Lcms**: 376 (M + H)

### *tert*-butyl 2-((2,6-dimethoxy-4-(4,4,5,5-tetramethyl-1,3,2-dioxaborolan-2-yl)benzyl)(methyl)amino)acetate

*tert*-butyl 2-((4-bromo-2,6-dimethoxybenzyl)(methyl)amino)acetate (300 mg, 0.802 mmol, 1.0 eq) and bis(pinacolato)diboron (305 mg, 1.20 mmol, 1.5 eq), were dissolved in DMF, before KOAC (394 mg, 4.01 mmol, 5.0 eq), and $PdCl_2$(dppf) • $CH_2Cl_2$ (65.5 mg, 0.080 mmol, 0.1 eq) were added. The mixture was degassed, and headspace flushed with $N_2$ before heating to 90°C for 16 hr. The reaction was diluted to 80 mL with EtOAc, filtered through celite, and washed twice with a 1:1 solution of deionized water and saturated brine (20 mL), three times with deionized water (20 mL), and once with saturated brine (20 mL), before being dried over $Na_2SO_4$ and concentrated in vacuo. The residue was dissolved in DCM and purified by silica chromatography (EtOAc/Hexanes 0% to 100% gradient) to give the desired product as a brown solid (158 mg, 47%).

**Chemical structure 3.** *tert*-butyl 2-((2,6-dimethoxy-4-(4,4,5,5-tetramethyl-1,3,2-dioxaborolan-2-yl)benzyl)(methyl)amino)acetate.
DOI: https://doi.org/10.7554/eLife.41305.034

**[1]H NMR:** (500 MHz, Chloroform-*d*) δ = 6.98 (s, 2H), 3.90 (s, 2H), 3.85 (s, 6H), 3.20 (s, 2H), 2.41 (s, 3H), 1.48 (s, 9H), 1.35 (s, 12H).
**Lcms:** 423 (M + H)

### tert-butyl 2-((2,6-dimethoxy-4-(2-methyl-1-oxo-1,2-dihydro-2,7-naphthyridin-4-yl)benzyl)(methyl)amino)acetate

4-bromo-2-methyl-2,7-naphthyridin-1(2H)-one (476 mg, 2.0 mmol, 1.0 eq) and tert-butyl 2-((2,6-dimethoxy-4-(4,4,5,5-tetramethyl-1,3,2-dioxaborolan-2-yl)benzyl)(methyl)amino)acetate (1.01 g, 2.4 mmol, 1.2 eq) (prepared over multiple batches as above) were dissolved in DMF (10 mL) before a 2N solution of Na2CO3 was added (2.5 mL, 5 mmol, 2.5 eq) followed by Pd(dppf)Cl2 • DCM (366 mg, 0.2 mmol, 0.1 eq). The mixture was degassed and heated to 80°C overnight. Solvent was removed by lyophilization and the crude product was used directly.

**Chemical structure 4.** *tert*-butyl 2-((2,6-dimethoxy-4-(2-methyl-1-oxo-1,2-dihydro-2,7-naphthyridin-4-yl)benzyl)(methyl)amino)acetate.
DOI: https://doi.org/10.7554/eLife.41305.035

**Lcms:** 454 (M + H)

## 2-((2,6-dimethoxy-4-(2-methyl-1-oxo-1,2-dihydro-2,7-naphthyridin-4-yl)benzyl)(methyl)amino)acetic acid

The above residue was dissolved in DCM (2 mL) before TFA (2 mL) was slowly added. After stirring at rt for 24 hr, the mixture was concentrated in vacuo. The residue was purified by prep-HPLC (0.05% TFA) to give the desired product as an off-white solid (410 mg, 51% over 2 steps)

**Chemical structure 5.** 2-((2,6-dimethoxy-4-(2-methyl-1-oxo-1,2-dihydro-2,7-naphthyridin-4-yl)benzyl)(methyl)amino) acetic acid.
DOI: https://doi.org/10.7554/eLife.41305.036

**1H NMR** (500 MHz, DMSO-d6) δ = 9.76 (s, 1H), 9.48 (s, 1H), 8.75 (d, 1H), 7.94 (s, 1H), 7.64 (d, 1H), 6.87 (s, 2H), 4.42 (s, 2H), 4.02 (s, 2H), 3.87 (s, 6H), 3.63 (s, 3H), 2.76 (s, 3H).
**Lcms:** 398 (M + H)

## 4-((8-aminooctyl)amino)−2-(2,6-dioxopiperidin-3-yl)isoindoline-1,3-dione (4)

To a solution of 2-(2,6-dioxopiperidin-3-yl)−4-fluoroisoindoline-1,3-dione (800 mg, 2.9 mmol) and tert-butyl (8-aminooctyl)carbamate (710 mg, 2.9 mmol) in NMP (15 mL, 0.2 M) was added DIPEA (451 mg, 3.5 mmol). The mixture was stirred at 90℃ overnight, cooled to room temperature, diluted with EtOAc (100 mL), and washed with water (3 × 50 mL). The organic phase was washed with brine (50 mL), dried over anhydrous Na2SO4, and filtered. The filtrate was concentrated in vacuo, and the residue was stirred in TFA/CH2 Cl2 (2 mL / 4 mL) for 2 hr at rt. The volatile was removed and the residue was purified by prep-HPLC (0.05% TFA in CH3CN/H2O) to afford the desired product (687 mg, 46%) as a yellow solid.

**Chemical structure 6.** 4-((8-aminooctyl)amino)-2-(2,6-dioxopiperidin-3-yl)isoindoline-1,3-dione (4).
DOI: https://doi.org/10.7554/eLife.41305.037

**1H NMR** (500 MHz, Methanol-d4) δ 7.59–7.51 (m, 1H), 7.04 (dd, J = 7.9, 1.7 Hz, 2H), 5.06 (dd, J = 12.4, 5.5 Hz, 1H), 3.34 (d, J = 7.0 Hz, 2H), 2.95–2.81 (m, 3H), 2.79–2.66 (m, 2H), 2.15–2.08 (m, 1H), 1.67 (tt, J = 12.2, 7.2 Hz, 4H), 1.43 (d, J = 22.2 Hz, 8H).
**Lcms** 401.39 (M + H)

## dBRD9-A

To a solution of 4-((8-aminooctyl)amino)−2-(2,6-dioxopiperidin-3-yl)isoindoline-1,3-dione trifluoroacetate salt (669 mg, 1.3 mmol) and 2-((2,6-dimethoxy-4-(2-methyl-1-oxo-1,2-dihydro-2,7-naphthyridin-4-yl)benzyl)(methyl)amino)acetic acid (520 mg, 1.3 mmol) in DMF (5 mL) was added HATU (990 mg, 2.6 mmol) and DIPEA (516 mg, 4 mmol). The mixture was stirred at rt. for 2 hr, diluted with ethyl acetate (50 mL), and washed with water (3 × 20 mL) and brine (20 mL), dried over anhydrous Na2SO4, filtered and concentrated. The residue was purified by prep-HPLC (0.05% TFA in CH3CN/H2O) to afford **dBRD9-A** (583 mg, 50%) as a yellow solid.

**Chemical structure 7.** dBRD9-A.
DOI: https://doi.org/10.7554/eLife.41305.038

$^{1}$H NMR (500 MHz, Methanol-$d_4$) δ 9.54 (s, 1H), 8.67 (d, $J$ = 6.1 Hz, 1H), 7.89 (s, 1H), 7.77 (d, $J$ = 6.0 Hz, 1H), 7.51 (dd, $J$ = 8.5, 7.2 Hz, 1H), 6.99 (dd, $J$ = 7.8, 2.2 Hz, 2H), 6.84 (s, 2H), 5.48 (s, 2H), 5.03 (dd, $J$ = 12.6, 5.5 Hz, 1H), 4.51 (d, $J$ = 4.9 Hz, 2H), 3.95 (s, 6H), 3.70 (s, 3H), 3.34 (s, 1H), 3.27 (t, $J$ = 6.9 Hz, 2H), 2.92 (s, 3H), 2.85 (ddd, $J$ = 17.5, 13.9, 5.2 Hz, 1H), 2.76–2.65 (m, 2H), 2.13–2.06 (m, 1H), 1.61 (p, $J$ = 6.9 Hz, 2H), 1.52–1.46 (m, 2H), 1.43–1.25 (m, 11H).
Lcms: 780.9 (M + H)

## Accession numbers

All next-generation sequencing datasets generated in association with this work have been deposited in the Gene Expression Omnibus (GEO) under accession number GSE113229 (https://www.ncbi.nlm.nih.gov/geo/query/acc.cgi?acc=GSE113229).

## Acknowledgements

We thank members of the Armstrong, Vakoc and Bradner laboratories for engaging discussions throughout the course of this work. We thank B Knoechel, R Isenhart, S Potdar and J Kloeber for assistance with next-generation sequencing experiments. We thank X Wang for technical assistance in relation to chromatin immunoprecipitation experiments. This work was supported by grants from the NCI (CA176745, CA066996, CA204915) and Alex's Lemonade Stand Foundation to SAA GLB was supported by an EMBO Long-Term Fellowship (ALTF-1235–2015) and an Irish Cancer Society Biomedical Research Fellowship (CRF18BRI). The Vermeulen lab is part of the Oncode Institute, which is partly funded by the Dutch Cancer Society (KWF). SAA is a consultant and/or shareholder for Imago Biosciences, Cyteir Therapeutics, C4 Therapeutics, Syros Pharmaceuticals, OxStem Oncology, ProQR and Accent Therapeutics. SAA receives research support from Janssen, Novartis, and AstraZeneca.

## Additional information

### Competing interests

Nathanael S Gray: is a founder, science advisory board member (SAB) and equity holder in Gatekeeper, Syros, Petra, C4, B2S and Soltego. The Gray lab receives or has received research funding from Novartis, Takeda, Astellas, Taiho, Janssen, Kinogen, Her2llc, Voronoi, Deerfield and Sanofi. James E Bradner: is now an executive and shareholder of Novartis AG, and has been a founder and shareholder of SHAPE (acquired by Medivir), Acetylon (acquired by Celgene), Tensha (acquired by Roche), Syros, Regency and C4 Therapeutics. Christopher R Vakoc: is an advisor to KSQ Therapeutics and receives research support from Boehringer-Ingelheim. Scott A Armstrong: is a consultant and/or shareholder for Imago Biosciences, Cyteir Therapeutics, C4 Therapeutics, Syros Pharmaceuticals, OxStem Oncology and Accent Therapeutics. SAA has received research support from Janssen, Novartis, and AstraZeneca. The other authors declare that no competing interests exist.

## Funding

| Funder | Grant reference number | Author |
|---|---|---|
| National Cancer Institute | CA176745 | Scott A Armstrong |
| Alex's Lemonade Stand Foundation for Childhood Cancer | | Scott A Armstrong |
| European Molecular Biology Organization | ALTF-1235-2015 | Gerard L Brien |
| Irish Cancer Society | CRF18BRI | Gerard L Brien |
| National Cancer Institute | CA066996 | Scott A Armstrong |
| National Cancer Institute | CA204915 | Scott A Armstrong |

The funders had no role in study design, data collection and interpretation, or the decision to submit the work for publication.

## Author contributions

Gerard L Brien, Conceptualization, Formal analysis, Funding acquisition, Methodology, Writing—original draft; David Remillard, Jonathon Chabon, Kieran Wynne, Gerard Cagney, Methodology; Junwei Shi, Conceptualization, Formal analysis, Methodology; Matthew L Hemming, Data curation; Eugène T Dillon, Guido Van Mierlo, Marijke P Baltissen, Michiel Vermeulen, Formal analysis, Methodology; Jun Qi, Resources, Methodology; Stefan Fröhling, Nathanael S Gray, Resources; James E Bradner, Resources, Supervision; Christopher R Vakoc, Conceptualization, Resources, Supervision; Scott A Armstrong, Conceptualization, Supervision, Funding acquisition

## Author ORCIDs

Gerard L Brien (iD) http://orcid.org/0000-0003-4275-7178
Nathanael S Gray (iD) https://orcid.org/0000-0001-5354-7403
Scott A Armstrong (iD) http://orcid.org/0000-0002-9099-4728

## Ethics

Animal experimentation: All mouse experiments were performed according to approved institutional animal care and use committee (IACUC) protocols at the Dana Farber Cancer Institute.

## Decision letter and Author response

Decision letter https://doi.org/10.7554/eLife.41305.043
Author response https://doi.org/10.7554/eLife.41305.044

# Additional files

## Supplementary files

• Transparent reporting form
DOI: https://doi.org/10.7554/eLife.41305.039

## Data availability

All next-generation sequencing datasets generated in association with this work have been deposited in the Gene Expression Omnibus (GEO) under accession number GSE113229

The following dataset was generated:

| Author(s) | Year | Dataset title | Dataset URL | Database and Identifier |
|---|---|---|---|---|
| Brien GL | 2018 | Targeted degradation of BRD9 reverses oncogenic gene expression in synovial sarcoma | https://www.ncbi.nlm.nih.gov/geo/query/acc.cgi?acc=GSE113229 | NCBI Gene Expression Omnibus, GSE113229 |

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
