## [Decision Letter]

Thank you for submitting your article "Targeted degradation of BRD9 reverses oncogenic gene expression in synovial sarcoma" for consideration by *eLife*. Your article has been reviewed by two peer reviewers, and the evaluation has been overseen by a Reviewing Editor and Charles Sawyers as the Senior Editor. The following individual involved in review of your submission has agreed to reveal her identity: Elizabeth Lawlor (Reviewer #2).

The reviewers have discussed the reviews with one another and the Reviewing Editor has drafted this decision to help you prepare a revised submission.

Summary:

This is an innovative and compelling paper that describes the discovery of BRD9 as a key dependency in synovial sarcoma. Furthermore, it defines two functional domains of the protein, including a novel non-bromodomain, that are essential for BRD9 function in SYS-SSX fusion protein biology and GBAF complex assembly. The paper is extremely well written and clear. The data are generally thorough and complete with appropriate controls and rigor. The results of the work raise many new questions related to biology of the BAF complexes that are now open for further study. In addition, they provide a platform to develop exciting more specific targeted therapeutics that could hold promise for synovial sarcoma therapy.

Essential revisions:

An outstanding question that arises is whether the impact of BRD9 inhibition/degradation is directly mediated by loss of BRD9 function itself or indirectly through effects on the fusion-bound GBAF complex, or indeed, the fusion protein itself. This question could be addressed by answering the following questions:

1) The authors speculate that BRD9 exists in complex with the fusion in the context of the newly described GBAF complex. This is worth further discussion in light of what Alpsoy and Dykhuizen recently published on the GBAF complex. Ideally, inclusion of co-IP data could show whether or not BRD9 associates with GBAF members and not with core BAF subunits, as described in Alpsoy and Dykhuizen (2018).

2) Does BRD9 inhibition alter the stability or composition of either the GBAF or core BAF complexes?

3) Further, is the stability/half-life of the fusion protein altered by either BRD9 inhibitor and/or degrader?

One additional point to be addressed is to clarify what proportion of fusion-regulated genes have altered expression following dBRD9-A treatment? In Figures 4 and 5, the authors show that despite >95% degradation of BRD9, a dose of 100nM compound results in only 50% growth inhibition. Figure 5F,G,H limits the analysis to target genes of the fusion that are regulated by super-enhancers. What happens to the fusion gene signature more broadly? Given the continued proliferation of most cells, even in vitro, one wonders if the fusion is still driving gene expression despite the relative loss of chromatin binding that was shown in Figure 4G.

---

## [Author Response]

Essential revisions:An outstanding question that arises is whether the impact of BRD9 inhibition/degradation is directly mediated by loss of BRD9 function itself or indirectly through effects on the fusion-bound GBAF complex, or indeed, the fusion protein itself. This question could be addressed by answering the following questions:1) The authors speculate that BRD9 exists in complex with the fusion in the context of the newly described GBAF complex. This is worth further discussion in light of what Alpsoy and Dykhuizen recently published on the GBAF complex. Ideally, inclusion of co-IP data could show whether or not BRD9 associates with GBAF members and not with core BAF subunits, as described in Alpsoy and Dykhuizen (2018).

This is a fair comment given that we did not demonstrate in our previous manuscript that BRD9 assembles into GBAF complexes; since our proteomics analyses focused exclusively on the fusion protein. We have attempted to perform co-IP western blots to understand whether BRD9 co-purifies GBAF members (GLTSCR1 and GLTSCR1L) in synovial sarcoma cells. However, in our hands current GLTSCR1/L antibodies do not facilitate consistent, quality western blotting which has been a limiting factor in this regard. However, in lieu of effective GLTSCR1/L western blotting reagents we have performed unbiased proteomics of fusion protein purifications in DMSO and dBRD9-A treated cells (Figure 4—figure supplement 1H). These analyses provide strong evidence for the incorporation of BRD9 into GBAF complexes in this setting. We find that degradation of BRD9 not only leads to a loss of BRD9 from SS18-SSX containing complexes; but also a loss of both GLTSCR1/L. Importantly, the abundance of other (non-GBAF) complex members is unaffected in these experiments indicating that BRD9 likely assembles into GBAF complexes. Moreover, these data suggest that incorporation of BRD9 into GBAF is essential for proper complex assembly as evidenced by the loss of GLTSCR1/L in the absence of BRD9.

2) Does BRD9 inhibition alter the stability or composition of either the GBAF or core BAF complexes?

This is an important question which our new proteomics data (described above) provides significant insights on. We find that BRD9 degradation leads to an at least partial disassembly of GBAF complexes (Figure 4—figure supplement 1H). We observe no other changes in the abundance of BAF members within SS18-SSX containing complexes. These findings suggest that degradation of BRD9 primarily effects GBAF complex assembly but does not appear to shift the dynamics of other non-GBAF forms of the complex.

3) Further, is the stability/half-life of the fusion protein altered by either BRD9 inhibitor and/or degrader?

This is an important point for clarification given that “off-target” degradation of the SS18-SSX fusion protein itself could account for the phenotypic effects observed in synovial sarcoma cells. However, western blotting experiments of inhibitor and degrader cells (Figure 4—figure supplement 1G) demonstrate that SS18-SSX levels are unchanged in inhibitor/degrader treated cells. Importantly, this supports the hypothesis that the GBAF assembly is essential in synovial sarcoma; and that the observed growth inhibition is not owing to collateral effects on the SS18-SSX fusion stability.

One additional point to be addressed is to clarify what proportion of fusion-regulated genes have altered expression following dBRD9-A treatment? In Figures 4 and 5, the authors show that despite >95% degradation of BRD9, a dose of 100nM compound results in only 50% growth inhibition. Figure 5F,G,H limits the analysis to target genes of the fusion that are regulated by super-enhancers. What happens to the fusion gene signature more broadly? Given the continued proliferation of most cells, even in vitro, one wonders if the fusion is still driving gene expression despite the relative loss of chromatin binding that was shown in Figure 4G.

This is a fair comment and we have extended our analyses in the revised manuscript to address this point (Figure 5—figure supplement 1G). BRD9 and SS18-SSX bind broadly throughout the genome and as such a large fraction of active genes are direct fusion protein targets. BRD9 degradation induces a relatively limited transcriptional response – 220 genes downregulated at 6hrs post-degradation. In a genome-wide context this translates to down-regulation of ~1.5% of direct SS18-SSX target genes. As such in a broad sense BRD9 targeting does not globally perturb the SS18-SSX gene expression signature. But rather impinges on a more specific subset of genes whose expression is driven by super enhancer elements. More generally these data suggest that GBAF complexes likely execute specific functions in the context of mSWI/SNF complexes generally. Highlighting the importance of additional functional studies to dissect the contribution of GBAF complexes in both normal and cancer cells.